# Orthogonal Cas9–Cas9 chimeras provide a versatile platform for genome editing

Mehmet Fatih Bolukbasi[1,2,6], Pengpeng Liu[1], Kevin Luk[1], Samantha F. Kwok[1], Ankit Gupta[1,7], Nadia Amrani[3], Erik J. Sontheimer [3,4], Lihua Julie Zhu [1,4,5] & Scot A. Wolfe[1,2]

The development of robust, versatile and accurate toolsets is critical to facilitate therapeutic genome editing applications. Here we establish RNA-programmable Cas9-Cas9 chimeras, in single- and dual-nuclease formats, as versatile genome engineering systems. In both of these formats, Cas9-Cas9 fusions display an expanded targeting repertoire and achieve highly specific genome editing. Dual-nuclease Cas9-Cas9 chimeras have distinct advantages over monomeric Cas9s including higher target site activity and the generation of predictable precise deletion products between their target sites. At a therapeutically relevant site within the *BCL11A* erythroid enhancer, Cas9-Cas9 nucleases produced precise deletions that comprised up to 97% of all sequence alterations. Thus Cas9-Cas9 chimeras represent an important tool that could be particularly valuable for therapeutic genome editing applications where a precise cleavage position and defined sequence end products are desirable.

[1] Department of Molecular, Cell and Cancer Biology, University of Massachusetts Medical School, Worcester, MA, USA. [2] Department of Biochemistry and Molecular Pharmacology, University of Massachusetts Medical School, Worcester, MA, USA. [3] RNA Therapeutics Institute, University of Massachusetts Medical School, Worcester, MA, USA. [4] Program in Molecular Medicine, University of Massachusetts Medical School, Worcester, MA, USA. [5] Program in Bioinformatics and Integrative Biology, University of Massachusetts Medical School, Worcester, MA, USA. [6] Present address: Exonics Therapeutics, Watertown, MA, USA. [7] Present address: Bluebird Bio., Cambridge, MA, USA. These authors contributed equally: Mehmet Fatih Bolukbasi, Pengpeng Liu. Correspondence and requests for materials should be addressed to S.A.W. (email: scot.wolfe@umassmed.edu)

The Class 2 CRISPR-Cas bacterial adaptive immune system has been used for a wide variety of applications since being repurposed for programmable genome editing and gene regulation[1,2]. The development of these tools for therapeutic genome editing applications is well underway with numerous investigations examining ex vivo and in vivo therapeutic approaches[3,4]. The Type II effector protein Cas9 from *Streptococcus pyogenes* (SpCas9) is one of the most widely used of these nucleases due to its robust activity and broad targeting range[5,6]. Target site recognition involves Cas9 binding to its PAM recognition element and complementarity of the complexed guide RNA with the neighboring genomic sequence[5,7–9]. Once fully engaged via R-loop formation, SpCas9 typically generates a blunt double-strand break (DSBs) at the target site[5]. In some instances the accuracy of wild type SpCas9 (SpCas9$^{WT}$) nuclease is imperfect, leading to cleavage of "off-target" sites within the genome[10–12]. The resulting collateral damage to the genome is undesirable for many therapeutic applications.

We and others have used protein and RNA engineering strategies to improve the specificity of SpCas9 for therapeutic genome editing applications[13,14]. We previously reported that a chimera between PAM-interaction attenuated SpCas9 (SpCas9$^{MT}$) and a programmable DNA-binding domain (pDBD; ZFP or TALE) enhances the targeting range and specificity of SpCas9[15]. In the SpCas9$^{MT}$-pDBD chimeras the pDBD provides an additional stage of target site licensing prior to cleavage. The first stage is mediated by pDBD recognition of a sequence downstream of the PAM. The increased effective concentration of SpCas9$^{MT}$ upon pDBD binding facilitates recognition of the PAM element and initiation of R-loop formation[7,15]. If sufficient complementarity exists between the sgRNA and the target site, cleavage of the DNA strands occurs. However, building functional pDBDs requires some level of expertise in pDBD assembly, which creates a barrier to this platform's adoption.

In this study, we assessed the feasibility of substituting the pDBD within the Cas9$^{MT}$-pDBD platform with an orthogonal Cas9 from either *N. meningiditis* (NmCas9)[16,17] or *S. aureus* (SaCas9)[18] to develop an entirely RNA-programmable nuclease platform spanning two linked Cas9 domains. We constructed these Cas9-Cas9 chimeras in both single-nuclease and dual-nuclease formats. In the single-nuclease format, the attenuated SpCas9$^{MT}$ domain is fused to a nuclease-dead NmCas9 or SaCas9, whose orthogonal guide is programmed to target a neighboring DNA sequence. Here the nuclease-dead Cas9 (dNmCas9 or dSaCas9) should act like a pDBD to deliver SpCas9$^{MT}$ to the target site, thereby permitting target site recognition through the increased effective concentration of the SpCas9$^{MT}$ nuclease (Fig. 1a). Similar to SpCas9$^{MT}$-pDBDs[15], SpCas9$^{MT}$-dNm/SaCas9 chimeras achieve a high-level of specificity as assessed via GUIDE-seq[12] and targeted amplicon sequencing. In the dual-nuclease format, both nucleases within the orthogonal Cas9–Cas9 fusions are active. We hypothesized that synchronous cleavage of the genome at two neighboring positions will primarily produce segmental deletions with defined junctions (referred to as precise deletions), as is observed to varying extents when independent nucleases are targeted to neighboring sites within a genome[6,19,20]. When programmed to target composite sites within the human genome, SpCas9$^{WT}$-Nm/SaCas9$^{WT}$ nuclease fusions produce a larger fraction of precise deletions, as high as 97% of all lesions, than a pair of independent Cas9 monomers used simultaneously. Similar to SpCas9-pDBD chimeras[15], Cas9–Cas9 fusions in both the single-nuclease and dual-nuclease format expand the targeting range of SpCas9 by allowing the recognition of suboptimal PAMs.

These dual nucleases should be particularly useful for the disruption of therapeutically relevant regulatory elements within a genome. In sickle cell disease, one proposed therapeutic approach is to induce the expression of the fetal γ-globin gene by deleting the GATA1-binding motif within the erythroid-lineage-specific regulatory element (enhancer +58 kb) of the *BCL11A* gene[21–25]. Here, we show that Cas9-Cas9 fusions programmed to target sites spanning the GATA1 element can delete this regulatory element with greater efficiency and accuracy than separate Cas9/sgRNA complexes.

## Results

**Single-nuclease Cas9 fusions facilitate accurate editing.** To identify binding site parameters necessary for functional Cas9–Cas9 fusion activity, we used a plasmid GFP reporter assay[26] to detect DSB formation. In this assay, two protospacers with optimal cognate PAM elements (for SpCas9[5,27] and NmCas9[16,28] or SpCas9 and SaCas9[18,29]) are arrayed in tandem in four configurations with various intervening spacings (Fig. 1b and Supplementary Fig. 1). These Cas9 target sites interrupt a GFP coding sequence. DNA cleavage within this region can excise the GFP-disrupting Cas9 target sites through single-strand annealing DNA repair, which provides a readout of nuclease activity[26]. In this assay, SpCas9$^{WT}$ displays robust nuclease activity while the R1335K mutant (SpCas9$^{MT3}$)[15] has substantially reduced activity. Fusion of nuclease-dead NmCas9 or SaCas9 to the C-terminus of SpCas9$^{MT3}$ via an unstructured 66 amino acid linker (SpCas9$^{MT3}$-dNmCas9 or SpCas9$^{MT3}$-dSaCas9) restores nuclease activity in configurations where the SpCas9 protospacer is upstream of the orthogonal Cas9 target site (Fig. 1b). Notably, C-terminal fusions of dNm/SaCas9 are more successful at restoring the loss of activity of SpCas9$^{MT3}$. N-terminal fusions of dNm/SaCas9 to SpCas9$^{MT3}$ (dNmCas9–SpCas9$^{MT3}$ or dSaCas9–SpCas9$^{MT3}$) were less active in the GFP reporter assay with the single linker configuration that we tested (Supplementary Fig. 1b).

In our pilot experiments at genomic target sites, Cas9–dCas9 fusions displayed a low level of editing activity. We reasoned that this could be due to poor nuclear localization of the fusion protein exacerbated by the less efficient expression of the Cas9–Cas9 fusion protein (Supplementary Fig. 1). Immunofluorescent imaging of the Cas9–Cas9 fusions indicated that the position and number of nuclear localization signals (NLSs) impact nuclear import efficiency of the Cas9–Cas9 fusions (Supplementary Fig. 1). NLSs within the linker between the two Cas9 domains appeared to have little function. Addition of NLSs at N and C termini achieved efficient nuclear localization of the Cas9–Cas9 fusions. We used this architecture for the remainder of this study.

We believe that the enhancement of SpCas9$^{MT3}$ nuclease activity in the context of the SpCas9$^{MT3}$–dSaCas9 fusions is due to its increased effective concentration near the dSaCas9-binding site. If this hypothesis is valid, nuclease activity should be dependent on the separation between the SpCas9 and SaCas9-binding sites. Defining the distance dependence of this behavior provides a framework for calculating the density of target sites genome-wide that are accessible to the Cas9–Cas9 fusions. To assess the distance constraints for the fusion protein, we examined the editing activity of SpCas9$^{MT3}$-dSaCas9 for a target site in the *AAVS1* locus as a function of the intervening distance between a common SpCas9 target site and a series of dSaCas9 target sites shifted progressively further away (Fig. 1c, Supplementary Fig. 2 and Supplementary Data 1). These data reveal an enhancement in SpCas9$^{MT3}$ activity in the fusion protein for a separation distance of <200 bp between the two binding sites. Beyond this distance the activity of the attenuated SpCas9$^{MT3}$ reverts to that of the unfused nuclease.

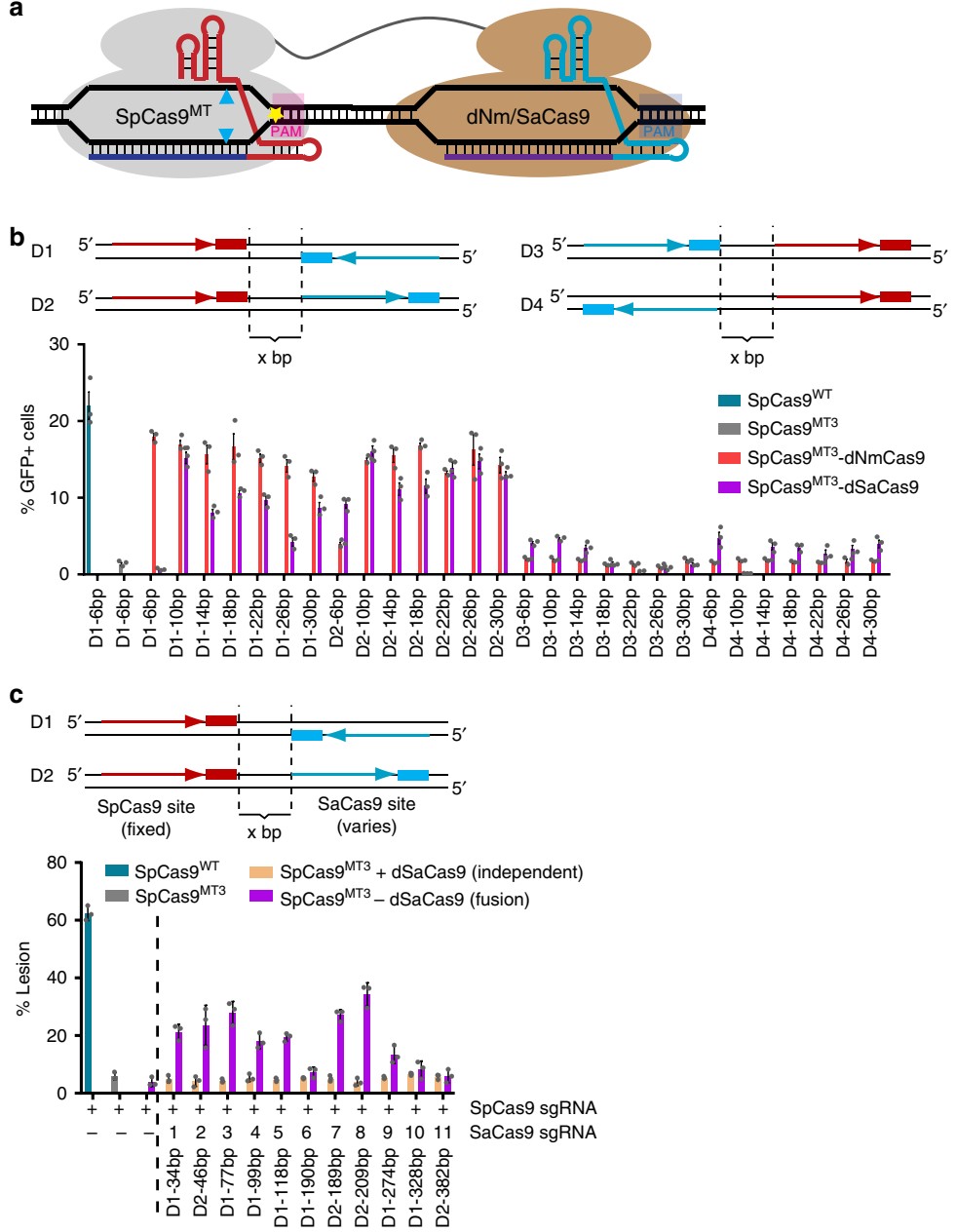

**Fig. 1** Development of a functional Cas9–Cas9 nuclease framework. **a** Schematic of SpCas9[MT]–dNm/SaCas9 fusions. PAM-interaction attenuated SpCas9[15] (yellow star) is C-terminally fused to a nuclease-dead Cas9 from *N. meningiditis* or *S. aureus*. Each Cas9 is loaded with its cognate sgRNA. **b** Top, schematic of parameters tested for target site organization. Four composite target site configurations are tested (D1:D4). The red arrow and rectangle represent the SpCas9 protospacer (in 5'–3' orientation) and PAM, respectively, whereas the blue arrow and rectangle represent the Nm/SaCas9 protospacer (in 5'–3' orientation) and PAM, respectively. Two dashed lines indicate the edges of each orthogonal Cas9-binding site, and x represents the number of intervening nucleotides. Bottom, activity profiles of SpCas9 (blue), SpCas9[MT3] (R1335K; gray), and C-terminal fusions for SpCas9[MT3]–dNmCas9 (pink), and SpCas9[MT3]-dSaCas9 (purple) in the GFP reporter assay. GFP reporter assay data are from three independent biological replicates performed on different days in HEK293T cells. **c** Changes in the activity profile of SpCas9[MT3]-dSaCas9 nuclease as a function of the distance between sgRNA-binding sites at the *AAVS1* locus. Top, schematic of the orientation of the target sites, where the SpCas9 site is fixed and the SaCas9 site is shifted away various distances to examine its distance-dependent activity (Supplementary Fig. 2) Bottom, deep sequencing data are from three independent biological replicates performed on different days in HEK293T cells, where the orientation and spacing between the orthogonal Cas9 sites is indicated below the *x*-axis (Supplementary Data 1). Error bars indicate ± s.e.m

One of the salient features of the SpCas9[MT]-pDBD chimera is the improved specificity of this platform relative to SpCas9[WT] (Ref 15.). To evaluate the specificity of Cas9–Cas9 fusions, we programmed SpCas9[MT3]–dNmCas9 and SpCas9[MT3]–dSaCas9 fusions to recognize a SpCas9 site (*VEGFA-TS3*)[10,12], which has numerous highly active off-target sites (Fig. 2a). At the *VEGFA-TS3* site, both SpCas9[MT3]–dNmCas9 and SpCas9[MT3]–dSaCas9 fusions display similar levels of on-target activity (Fig. 2b and Supplementary Data 1). We used GUIDE-seq[12] to assess the genome-wide specificity of these nucleases. In comparison to

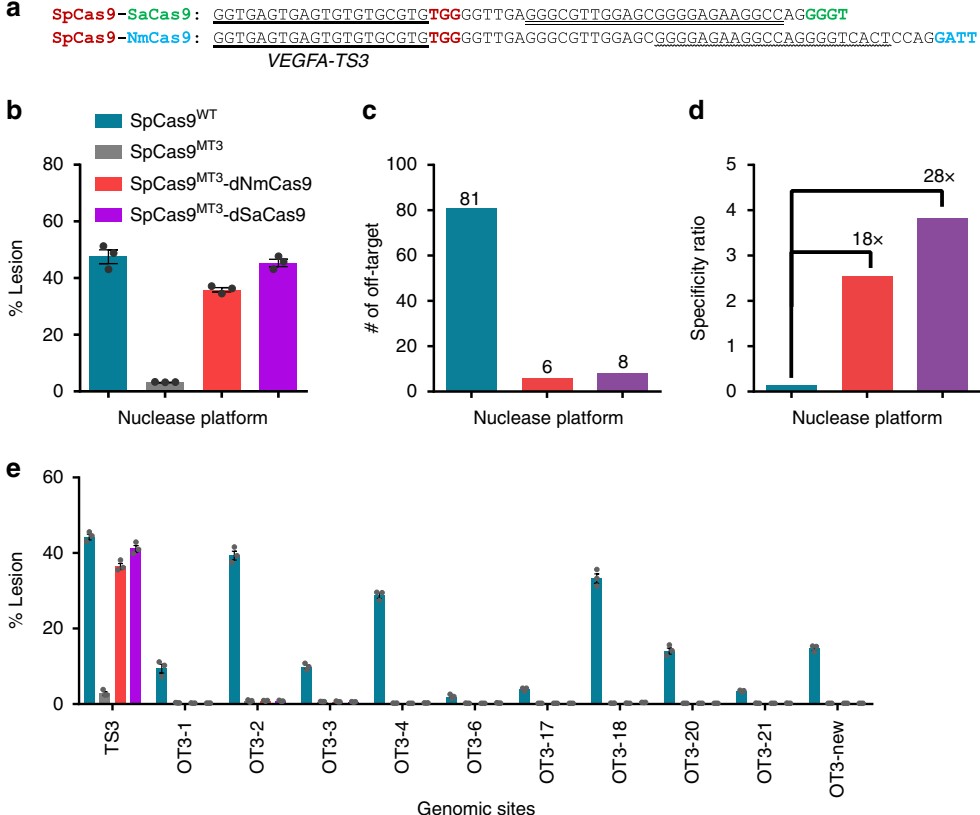

**Fig. 2** SpCas9$^{MT}$–dSa/NmCas9 fusions improve the specificity of editing. **a** Sequences of dual Cas9 genomic target sites at the *VEGFA* locus. The SpCas9 protospacer is bold underlined with its PAM is in red, the SaCas9 protospacer is double underlined with its PAM is green, and the NmCas9 protospacer is wavy underlined with its PAM in blue. **b** Lesion rates of the nuclease platforms are determined by deep sequencing. **c**, **d** Genome-wide off-target analysis of the nuclease platforms determined via GUIDE-seq[12] (Supplementary Data 2). **c** The number of off-target peaks detected for the given nuclease. **d** Fold improvement of the specificity ratio of the Cas9$^{MT}$–dCas9 framework relative to SpCas9$^{WT}$. **e** Deep sequencing determined lesion rates of the nucleases at a small set of off-target sites discovered within the GUIDE-seq data. GUIDE-seq result is from a single experiment, whereas amplicon deep sequencing data are from three independent biological replicates performed on different days in HEK293T cells (Supplementary Data 1). Error bars indicate ± s.e.m

SpCas9$^{WT}$, both SpCas9$^{MT3}$–dNmCas9 and SpCas9$^{MT3}$–dSaCas9 fusions have a substantially reduced number of GUIDE-seq peaks (Fig. 2c and Supplementary Data 2). In addition to the reduction in the numbers of GUIDE-seq peaks, the Specificity Ratios (number of unique capture events at the target site divided by sum of unique capture events at all off-target sites) for the SpCas9$^{MT3}$–dNmCas9 and SpCas9$^{MT3}$–dSaCas9 fusions are higher than for SpCas9$^{WT}$ (Fig. 2d and Supplementary Data 2). To validate the off-target editing rates, we used targeted amplicon deep sequencing for a representative set of off-target sites within the genomes of the treated cells. Despite being comparably active to SpCas9$^{WT}$ at the target site, SpCas9$^{MT3}$–dNmCas9 and SpCas9$^{MT3}$–dSaCas9 fusions present dramatically better discrimination against near-cognate off-target sites (Fig. 2e and Supplementary Data 1). Overall, GUIDE-seq and targeted amplicon deep-sequencing datasets indicate that SpCas9$^{MT}$–dNm/SaCas9 fusions substantially improve the specificity of SpCas9$^{WT}$ similar to the levels of achieved by SpCas9-pDBDs[15].

**Cas9-Cas9 dual nucleases generate precise deletions**. Unlike SpCas9-pDBDs, the Cas9–Cas9 fusions can also be used as dual nucleases. We hypothesized that when a pair of Cas9 nucleases dock together on a target site, they will generate two DSBs synchronously. Since rejoining of the broken ends without resection via canonical NHEJ (cNHEJ) is the predominant DSB repair

response in mammalian cells[30], the primary editing outcome of Cas9–Cas9 dual nucleases should be the precise deletion of the intervening segment between the cleavage sites. Previous studies have demonstrated that wild-type Cas9 nucleases that are targeted to a pair of neighboring genomic sequences can produce precise deletions but with variable efficiencies[6,19,20]. Such variability in editing outcomes is possibly due to asynchronous cleavage arising from differences in the efficiency of recognition or cleavage by the nuclease at two different sequences. Cas9–Cas9 dual nucleases should improve the level of synchrony by delivering both nucleases simultaneously to a pair of target sequences.

To test this hypothesis, we programmed SpCas9$^{WT}$-NmCas9$^{WT}$ dual nucleases to target the *VEGFA* locus in HEK293T cells. PCR amplification of the *VEGFA* locus from nuclease-treated cells indicates that dual-Cas9 nucleases, either independently or as fusions, generate precise segmental deletions, as anticipated (Fig. 3a, Supplementary Fig. 3). In principle, two DSBs at a nearby locus in the genome can produce at least six possible repair outcomes: random indels at the first nuclease cut site (SpCas9 indel), random indels at the second nuclease cut site (NmCas9 indel), random indels at both nuclease cut sites (Sp&NmCas9 indel), precise deletions, imprecise deletions, and inversions. To monitor the presence and distribution of each set of repair events quantitatively, we applied targeted amplicon deep sequencing of the genomic DNAs from cells treated with the different nuclease platforms. Analysis of the *VEGFA* amplicon sequencing data

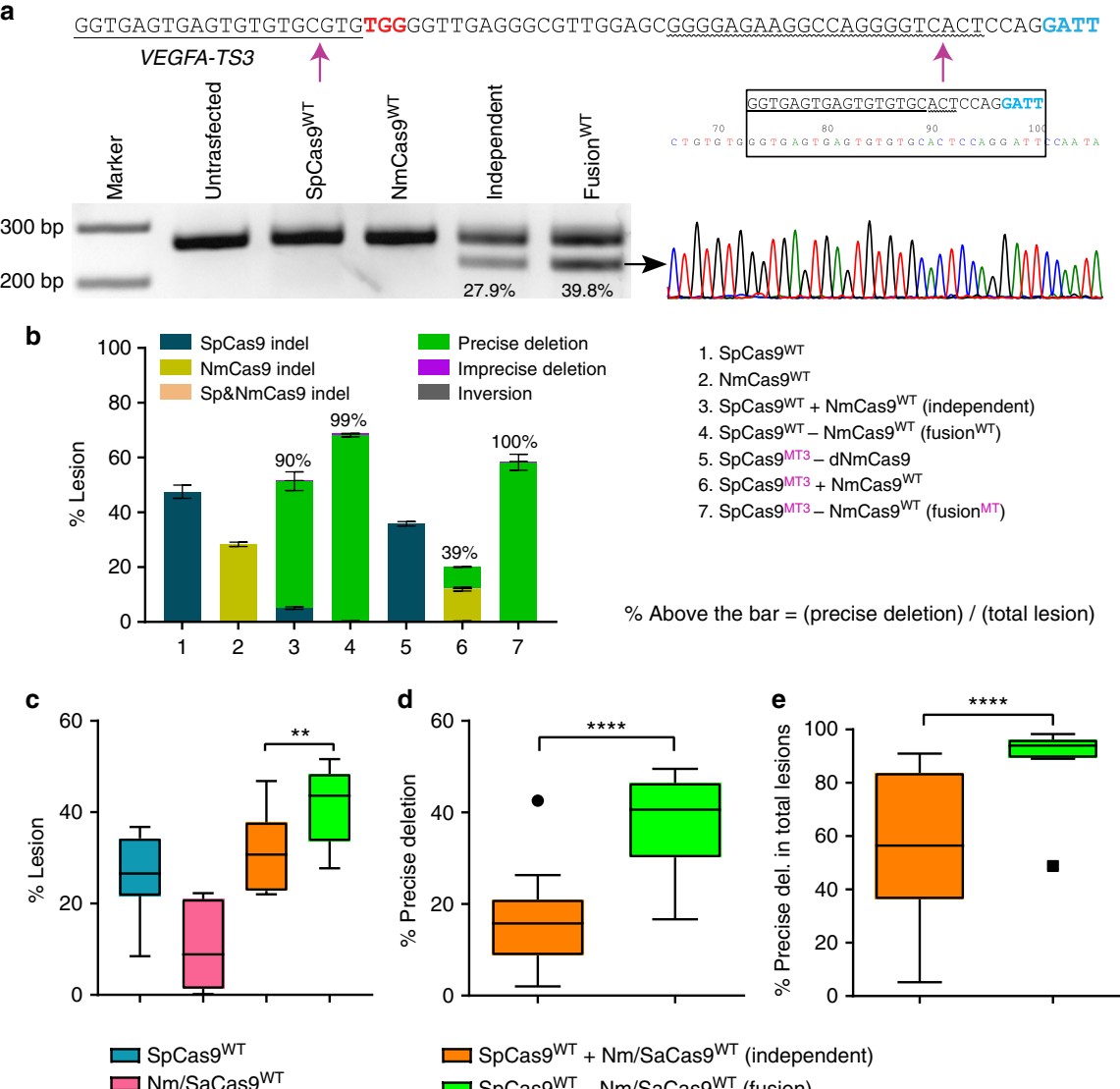

**Fig. 3** Cas9–Cas9 dual nucleases generate uniform deletion products. **a** Top, sequence of SpCas9-NmCas9 target site: SpCas9 protospacer is underlined and its PAM is red; NmCas9 protospacer is wavy underlined and its PAM is blue. Purple arrows indicate expected double-strand break positions. Bottom left, genomic region containing the target site is PCR amplified; the higher band is the wild-type sequence or sequences with small indels, and the lower band is the segmental deletion product generated by dual nucleases. Bottom right, chromatogram from Sanger sequencing of the gel-extracted lower band (black arrow). The main product is the perfect junction of two double-strand break sites yielding a precise deletion (black rectangle). **b** Lesion rates and types are determined by deep sequencing. Single nucleases generate small indels at their cleavage sites, whereas dual nucleases (independent or fusion) may generate six types of lesion products. The majority of the lesions produced by dual-nuclease fusions is precise deletion. SpCas9$^{MT}$–dNmCas9 fusions behave like a monomeric SpCas9. Values above each bar indicate the precise deletion rate divided by the total lesions. **c–e** Activity profiles of SpCas9$^{WT}$ (blue), Nm/SaCas9$^{WT}$ (pink), SpCas9$^{WT}$ + Nm/SaCas9$^{WT}$ (orange), and SpCas9$^{WT}$ − Nm/SaCas9$^{WT}$ (green) nucleases at 12 genomic sites (six D1 and 6 D2 configurations) determined by deep sequencing. **c** Total lesion rates of the Cas9–Cas9 dual nucleases are higher than the monomeric Cas9s used in combination. **d** Cas9–Cas9 fusions generate higher rates of precise deletions than two independent Cas9 monomers. **e** Cas9–Cas9 dual nucleases primarily generate precise deletion products whereas lesion types of the two independent monomeric Cas9s are site dependent. Each box plot is drawn by GraphPad Prism, where the box represents the 25th and 75th percentile and the middle line is the median. Whiskers and outliers are defined by the Tukey method. Statistical significance is determined by one-way analysis of variance (ANOVA), ** and **** denote $P < 0.01$ and $P < 0.0001$, respectively. Deep sequencing data are from three independent biological replicates performed on different days in HEK293T cells (Supplementary Data 1). Error bars indicate ± s.e.m

indicates two superior features of Cas9–Cas9 dual nuclease fusions (SpCas9$^{WT}$–NmCas9$^{WT}$, SpCas9$^{MT}$–NmCas9$^{WT}$) over two independent Cas9s (SpCas9$^{WT}$ + NmCas9$^{WT}$): Cas9–Cas9 fusions display higher levels of genome modification and a higher proportion of these editing events are precise deletions than are observed for two independent Cas9 nucleases (Fig. 3b and Supplementary Data 1). We observed similar editing outcomes for

SpCas9 and SaCas9 dual nucleases (Supplementary Fig. 4 and Supplementary Data 1).

To test the generality of this phenomenon, we screened an additional 41 genomic sites for the activity profiles of single and dual nucleases, and for the types of lesions that are produced within the genome. Overall, SpCas9 has a higher median nuclease activity than SaCas9 and NmCas9. The lesion rates of the dual

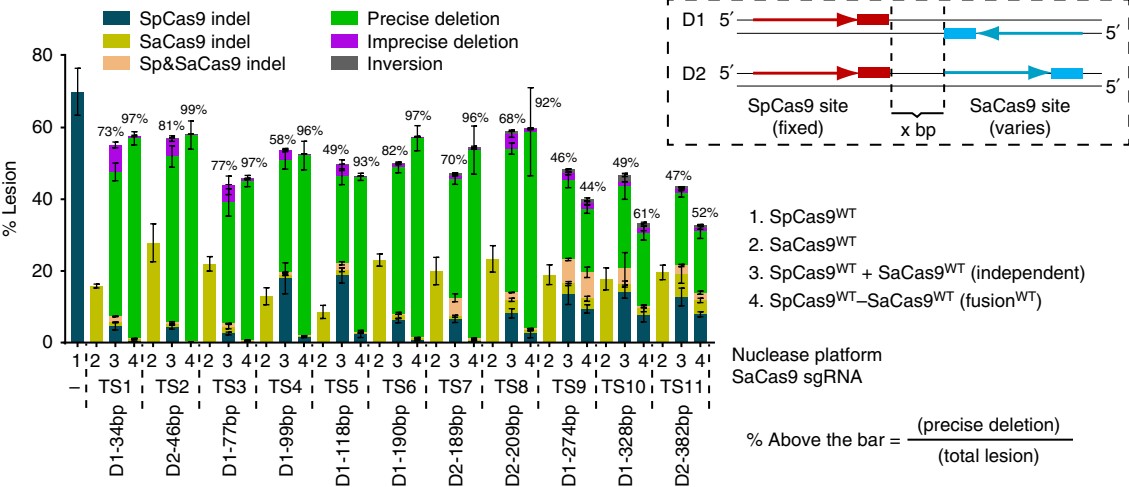

**Fig. 4** Cas9–Cas9 dual nucleases display enhanced precise deletions out to ~200 bp site separation. Changes in the activity profile of SpCas9WT–SaCas9WT nuclease as a function of the distance between sgRNA-binding sites at the *AAVS1* locus. Boxed inset: schematic of the orientation of the target sites, where the SpCas9 site is fixed and the SaCas9 site is shifted to examine the distance-dependent activity of the Cas9–Cas9 fusions (Supplementary Fig. 2). Bar graph: deep sequencing data, where the various lesion types and rates at *AAVS1* sites are determined by UMI-corrected deep sequencing. Data are from three independent biological replicates performed on different days in HEK293T cells, where the orientation and spacing between the orthogonal Cas9 sites is indicated below the *x*-axis (Supplementary Data 1). Error bars indicate ± s.e.m

independent nuclease are similar to the levels achieved by SpCas9 alone, but the composition of editing outcomes differs due to the production of precise deletions (Supplementary Fig. 5 and Supplementary Data 1). In comparison to dual independent nucleases, Cas9–Cas9 dual nucleases not only have higher overall activity, but also produce precise segmental deletions more efficiently (Supplementary Fig. 5 and Supplementary Data 1).

Next, we selected a representative set of 12 sites from these 41 sites that span different activity profiles for the individual nucleases to assess in greater depth the nuclease activities of single and dual nucleases for wild-type and PAM-interaction-deficient[15] forms. SpCas9WT–Sa/NmCas9WT dual-nuclease fusions have higher overall activity than SpCas9 and two independent Cas9s (Fig. 3c and Supplementary Data 1). This enhancement is likely due to cooperativity between the fused nucleases, where strong binding of one of the Cas9s increases the effective concentration and consequently the activity of the other nuclease. The total activity levels of SpCas9MT–Sa/NmCas9WT fusions are associated with the activity levels of the orthogonal Cas9:sgRNA complexes (Supplementary Fig. 6 and Supplementary Data 1). SpCas9WT–Sa/NmCas9WT dual-nuclease fusions double the production of precise deletions relative to two independent Cas9s (Fig. 3d and Supplementary Data 1). More importantly, precise deletions are the predominant products of Cas9–Cas9 dual nucleases, encompassing on average >90% of all lesion types (Fig. 3e, Supplementary Fig. 6 and Supplementary Data 1).

To determine the distance dependence for the synergistic deletion of sequence elements by Cas9–Cas9 fusions, we compared the activity of the SpCas9WT–SaCas9WT nuclease fusions to the independent nucleases at the sites with progressively greater separation between the Cas9-binding sites at the *AAVS1* locus (Supplementary Fig. 2). Similar to the data obtained for the SpCas9MT3–dSaCas9 fusions at this locus, we observed significantly higher rates of precise deletions for the dual nuclease fusions compared to the independent nucleases up to a separation of ~200 bp between the Cas9 target sites (Fig. 4, Supplementary Fig. 7, and Supplementary Data 1). Beyond this 200 bp threshold the fusion proteins behave like independent nucleases based on the decreased rates of precise deletions. These data suggest that generation of two

synchronous nearby breaks by Cas9–Cas9 dual nucleases are preferentially repaired via cNHEJ, and that the fusion proteins can function synergistically over modest distances within the genome.

**Defining the targeting range of the Cas9–Cas9 fusions.** Another salient feature of SpCas9–pDBD fusions is their increased targeting range achieved through their functionality at suboptimal PAM sequences[15]. To examine the targeting range of Cas9–Cas9 fusions, we designed several SpCas9 guides that target protospacers with suboptimal PAMs in tandem with an SaCas9 target site with an optimal N₂GRRT PAM sequence[18] (Supplementary Fig. 8). Wild-type SpCas9 has very low or no activity on these sites, as expected (Fig. 5a and Supplementary Data 1). However, in single-nuclease and dual-nuclease formats, SpCas9–SaCas9 fusions display nuclease activity at the SpCas9 sites with NAG, NTG, NCG, NGA, NGT, and NGC suboptimal PAMs. These data reflect the ability of the SpCas9–SaCas9 fusions to utilize the presence of a single guanine in the SpCas9 PAM element as a functional cleavage site (Fig. 5a and Supplementary Data 1).

One of the challenges to employing the Cas9–Cas9 system for targeted genome editing is the density of potential target sites given the requirement for two different PAM sequences. This problem is partially mitigated by the flexibility of the target site orientation and spacing for Cas9–Cas9 fusions (Fig. 1b, c), the ability to employ different orthogonal nucleases (SaCas9 or NmCas9), and the ability for SpCas9 to utilize alternate PAM sequences in the context of the fusion. We computationally estimated the number of potential target sites for both SpCas9–SaCas9 and SpCas9–NmCas9 fusions within the human genome based on two different criteria: optimal PAMs for both nucleases and conservative distance constraints (30 bp maximum separation; "canonical targets"), or alternate PAM sequences for SpCas9 with an optimal Sa/NmCas9 PAM and relaxed distance constraints (100 bp maximum separation; "expanded range"). Reassuringly, the conservative parameters indicate that SpCas9–SaCas9 fusions have about three-fold fewer target sites than SpCas9, and the relaxed parameters indicate that SpCas9–SaCas9 fusions have 2.5-fold more target sites than SpCas9 (Supplementary Fig. 8 and Supplementary Data 4). There

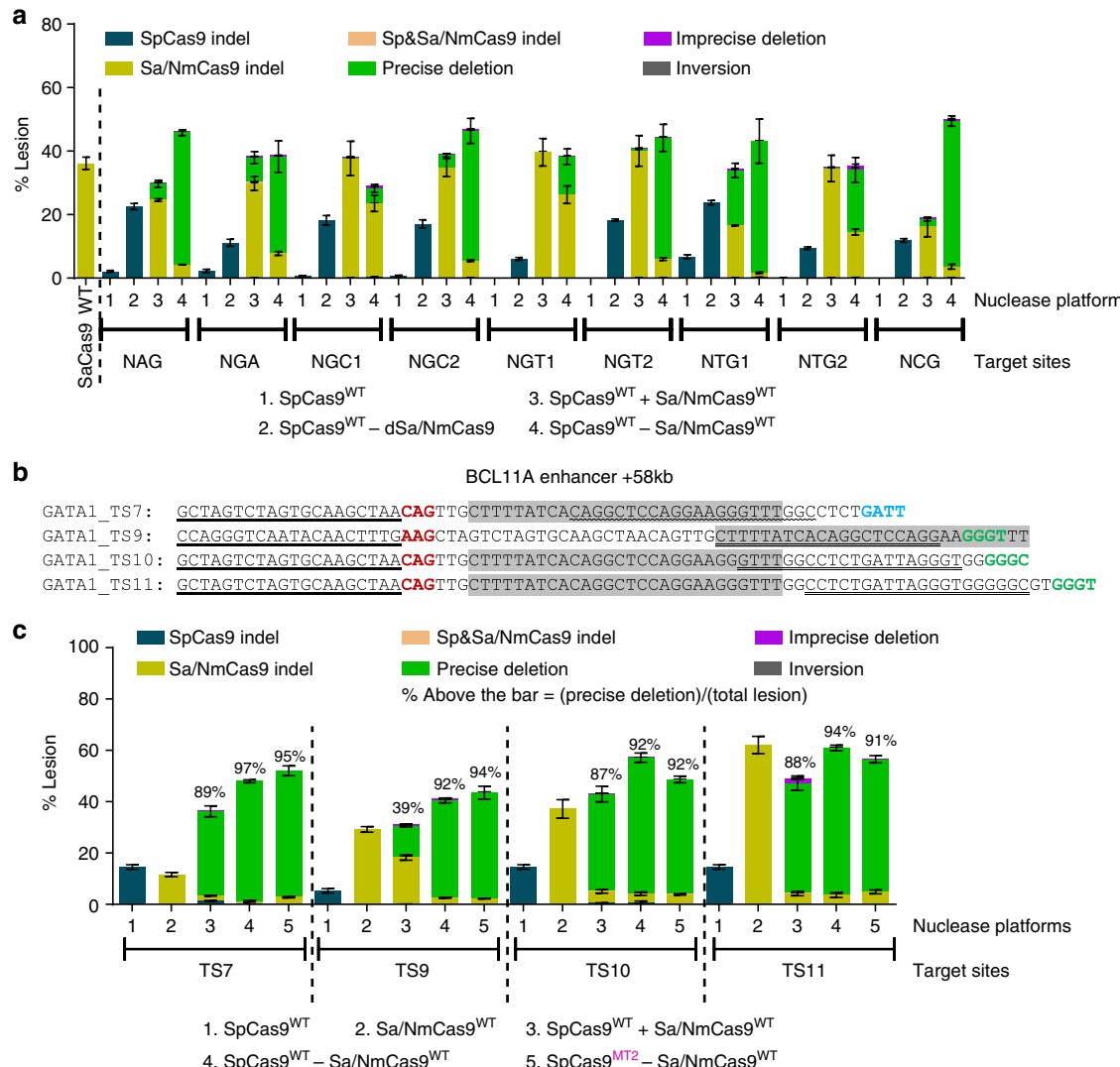

**Fig. 5** Cas9–Cas9 fusions expand the targeting range of SpCas9 allowing deletion of the GATA1-binding element in the *BCL11A* enhancer+58 kb. **a** Lesion rates and types at tandem target sites, with suboptimal SpCas9 but canonical SaCas9 PAMs (Supplementary Fig. 8) are determined by deep sequencing with bulk analysis. SaCas9 generates robust editing whereas SpCas9 displays low or no activity. In Cas9–Cas9 fusion format, SpCas9 cuts effectively at these protospacers as observed from the SpCas9–dSaCas9 fusions or the fused wild-type nucleases. **b** Sequence information of the four target sites chosen for more detailed assessment of the application of Cas9–Cas9 fusions for the deletion of the GATA1-binding element in the functional core of the *BCL11A* enhancer+58 kb (highlighted in gray[21]). **c** Lesion rates and types at four target sites spanning the GATA1 element are determined by deep sequencing after UMI-correction. Deep sequencing data are from three independent biological replicates performed on different days in HEK293T cells (Supplementary Data 1). Error bars indicate ± s.e.m

are fewer targetable sites for SpCas9–NmCas9 fusions, due to the more restrictive NmCas9 PAM requirement, but the number of predicted SpCas9–NmCas9 target sites under the relaxed parameters is similar to that for SpCas9.

**Cas9–Cas9 nucleases accurately delete regulatory elements.** Increased nuclease activity, enhanced targeting range, and the generation of uniform editing products favor the Cas9–Cas9 dual nucleases for the disruption of a gene or a regulatory element within genome. To apply this platform at a therapeutically relevant genomic locus, we tested the ability of Cas9–Cas9 dual nucleases to delete the GATA1-binding motif within the *BCL11A* erythroid-lineage-specific enhancer (+58 kb) element[21,22]. Disruption of this regulatory element in CD34+ hematopoietic stem and progenitor cells (HSPCs) silences *BCL11A* expression in the erythroid lineage, and thereby increases production of fetal γ-

globin protein in differentiated red blood cells[23]. Ex vivo genome editing of this locus in HSPCs in conjunction with autologous bone marrow transplantation is a potential therapeutic approach for the treatment for sickle cell disease[24,25].

To efficiently delete the GATA1-binding motif, we programmed Cas9–Cas9 fusions to target 12 different sites spanning this regulatory element in HEK293T cells (Supplementary Fig. 9). Similar to our analysis at other genomic loci, Cas9–Cas9 fusions effectively generate precise segmental deletions at most but not all of the examined sites (Supplementary Fig. 10 and Supplementary Data 1). Notably, in SpCas9^WT–Sa/NmCas9^WT dual-nuclease fusion format, all three Cas9s (Sp/Nm/SaCas9s) effectively cut protospacers with suboptimal PAM sequences: SpCas9 at NAG PAMs[27] (GATA1-TS7, TS8, TS9, TS10, and TS11) and GATA1-TS5 and TS9), NmCas9 at N4GCTT (GATA1-TS4), N4GTTT (GATA1-TS5), and N4GACT (GATA1-TS6) PAMs[16,28], SaCas9 at N2GGGC (GATA1-TS10), and N2GAGG (GATA1-TS12)

**a**

| Target site (TS) | Nuclease platform | Number of off-target sites | Number of unique reads at TS | Sum of unique reads at off-target sites |
|---|---|---|---|---|
| GATA1_TS7 | SpCas9$^{WT}$ | 0 | 79 | 0 |
| | NmCas9$^{WT}$ | 3 | 256 | 47 |
| | SpCas9$^{WT}$-NmCas9$^{WT}$ | 3 | 177 | 14 |
| GATA1_TS9 | SpCas9$^{WT}$ | 0 | 0 | 0 |
| | SaCas9$^{WT}$ | 0 | 1704 | 0 |
| | SpCas9$^{WT}$-SaCas9$^{WT}$ | 5 | 1809 | 47 |
| GATA1_TS10 | SpCas9$^{WT}$ | 0 | 79 | 0 |
| | SaCas9$^{WT}$ | 0 | 662 | 0 |
| | SpCas9$^{WT}$-SaCas9$^{WT}$ | 0 | 116 | 0 |
| GATA1_TS11 | SpCas9$^{WT}$ | 0 | 79 | 0 |
| | SaCas9$^{WT}$ | 43 | 1248 | 1938 |
| | SpCas9$^{WT}$-SaCas9$^{WT}$ | 17 | 210 | 388 |

**b**

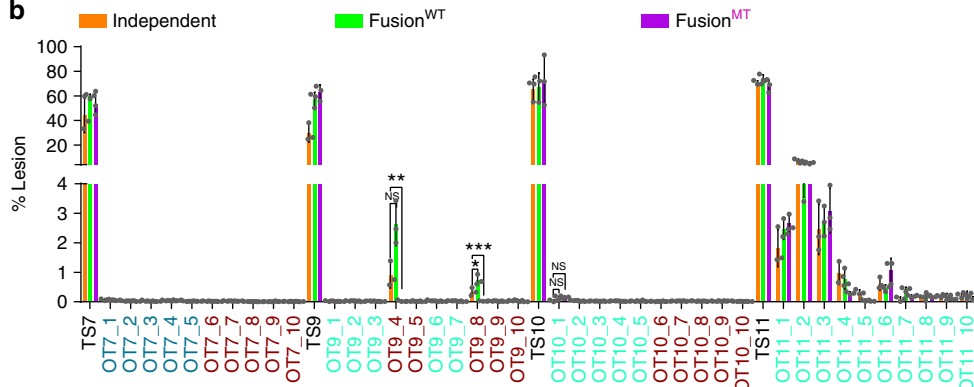

**Fig. 6** Cas9–Cas9 fusions achieve robust and specific genome editing. **a** Summary of the GUIDE-seq genome-wide off-target analysis of SpCas9$^{WT}$, Sa/NmCas9$^{WT}$, and SpCas9$^{WT}$-Sa/NmCas9$^{WT}$ at four GATA1 target sites (Supplementary Data 2). **b** Deep sequencing determined lesion rates for these nucleases at a subset of off-target sites discovered by the GUIDE-seq data or predicted by CasOFFinder (Supplementary Data 3). The names of SpCas9, NmCas9, and SaCas9 off-target sites are colored in dark red, blue, and green. The GUIDE-seq result is from single experiment, and amplicon deep sequencing data are from three independent biological replicates performed on different days in HEK293T cells (Supplementary Data 1). Error bars indicate ± s.e.m. Statistical significance is determined by one-way analysis of variance (ANOVA), *, **, ***, and NS denote BH-adjusted *P*-values of <0.05, <0.01, <0.001, and not significant, respectively

PAMs[18,29]. Among the 12 sites, we focused on four with the most promising activity, GATA1-TS7, GATA1-TS9, GATA1-TS10, and GATA1-TS11, for further characterization and specificity analysis (Fig. 5b and Supplementary Data 1).

One potential caveat for the accurate quantification of the lesion products produced at the target site is the PCR amplification bias of a precise deletion product due to its shorter length[31]. To address this possibility, we developed an assay based on unique-molecular identifiers (UMI)[32] and linear amplification-mediated (LAM) PCR[33] (Supplementary Fig. 11). In this approach, the genomic DNA is pre-amplified linearly with a single primer that is locus-specific, and that contains UMI and non-cognate adaptor sequences. Next, the UMI-containing single-stranded DNA is selectively amplified and barcoded for deep sequencing. We used the UMI-correction method to measure the lesion profiles of four GATA1 target sites within the *BCL11A* enhancer+58 kb locus. For single nucleases, there is no significant difference in the total lesion levels between bulk and UMI-corrected analyses. For dual nucleases, there is slight overestimation of the precise deletion levels in bulk sequencing data relative to the UMI-corrected analysis (Supplementary Fig. 11 and Supplementary Data 1).

Since many of the SpCas9 target sites in the *BCL11A* locus have suboptimal NAG PAMs, the activity levels of monomeric SpCas9

are modest at these target sites. Cas9–Cas9 dual nucleases display increased activity at all of these target sites relative to the dual independent nucleases. Notably, Cas9–Cas9 fusions containing the attenuated R1333S SpCas9 mutant (SpCas9$^{MT2}$)[15] also retain high activity at all four target sites with NAG PAMs, concordant with our previous findings (Fig. 5c and Supplementary Data 1). Similar to previous observations at other sites, even after UMI-correction, up to 97% of all lesions generated by Cas9–Cas9 dual nucleases are precise deletions (Fig. 5c and Supplementary Data 1). We performed activity assessments for the Cas9–Cas9 fusions at the GATA1-TS9 site in Jurkat and K562 cells, where we obtained similar relative activities and distributions of editing end-products as were observed in HEK293T cells (Supplementary Fig. 12). These data suggest that Cas9–Cas9 dual nucleases are a promising platform for the generation of uniform editing products at therapeutically important sites.

To evaluate the targeting specificity of these nucleases at the four GATA1 target sites, we performed GUIDE-seq genome-wide specificity analysis[12]. We screened for active off-target sites for each individual wild-type Cas9 and the wild type Cas9–Cas9 dual nucleases. We observed robust GUIDE-seq oligonucleotide incorporation at the target sites for all of the wild-type Cas9–Cas9 dual nucleases (Fig. 6a). A small number of potential off-target sites were identified for the TS7 and TS9 target sites based on the

GUIDE-seq analysis. A larger number of off-target sites were associated with the SaCas9 guide for the TS11 target site. The number of off-target sites identified by GUIDE-seq and the unique counts that are associated with each site are similar between the Cas9–Cas9 fusions and the individual Cas9 nucleases (Supplementary Data 2), suggesting that there is not a dramatic difference in the off-target activity between the individual nucleases and the Cas9–Cas9 fusions. The three off-target sites identified for NmCas9 have nine-nucleotide mismatches to the guide sequence, and are so divergent that they potentially represent false-positive sequences.

To assess the editing rate at potential off-target sites, we performed amplicon deep sequencing at these regions within the genome. We evaluated ten potential off-target sites for each nuclease, which were identified by either the GUIDE-seq analysis[12] or computationally predicted via Cas-OFFinder[34] (for TS7, TS9, and TS10) when insufficient high-quality sites were identified by GUIDE-seq (Supplementary Data 3). We evaluated the off-target activities of two different versions of the Cas9–Cas9 dual-nuclease fusions, SpCas9$^{WT}$–Sa/NmCas9$^{WT}$ or SpCas9$^{MT2}$–Sa/NmCas9$^{WT}$, in comparison with the independent SpCas9$^{WT}$ and Sa/NmCas9$^{WT}$ nucleases delivered simultaneously. The deep sequencing analysis indicates that the Cas9–Cas9 dual nuclease fusions display similar levels of off-target activity when compared to independent dual-Cas9 nucleases. Very low levels of mutagenesis (potentially background-level sequencing errors) were observed at the TS7 and TS10 off-target sites. More substantial mutagenesis was observed at two of the TS9 and many of the TS11 off-target sites. At two of the SpCas9 off-target sites (OT9-4 and OT9-8), the use of the attenuated SpCas9$^{MT2}$-SaCas9$^{WT}$ nuclease dramatically reduces the off-target activity compared to the independent wild-type SpCas9 or the SpCas9$^{WT}$–SaCas9$^{WT}$ nuclease, without a loss of on-target activity (Fig. 6b). These data demonstrate that Cas9–Cas9 dual nucleases achieve robust editing at the target site without generating a new class of off-target sites. The generation of uniform, accurate editing products within the *BCL11A* enhancer +58 kb locus highlights the utility of Cas9–Cas9 dual nucleases as a promising genome editing platform for the deletion of therapeutically relevant regulatory elements[24,25].

## Discussion

In this study, we have expanded the CRISPR/Cas9 toolset by developing orthogonal Cas9–Cas9 chimeras in single-nuclease and dual-nuclease formats. Unlike the original Cas9-pDBD platform, which required some expertise in protein engineering, the entirely RNA-programmable Cas9–Cas9 fusions should be completely accessible to the broader scientific community. In both single-nuclease and dual-nuclease formats, Cas9–Cas9 fusions act similarly to SpCas9-pDBDs with regards to enhanced targeting range and improved specificity[15]. The presence of a single guanine within the PAM is sufficient for SpCas9 to cleave its targets in the context of Cas9–Cas9 fusions. In principle, similar suboptimal PAM usage should be applicable for SaCas9 and NmCas9 when the SpCas9 targets an optimal NGG PAM element. For homology-directed repair applications, the ability to target sites with suboptimal PAM elements may allow Cas9–dCas9 fusions to generate a DSB closer to the site of the desired sequence conversion[35]. This feature can also be useful for allele-specific targeting by Cas9–dCas9 nucleases by placing the PAM recognition at a polymorphic site. In the dual-nuclease format, SpCas9$^{WT}$–Sa/NmCas9$^{WT}$ fusions display superior nuclease activity and primarily produce uniform, predictable lesions within the targeted genome. These fusions are also compatible with PAM-interaction-attenuated SpCas9 to provide an additional level of target site specificity. Finally, we believe that the Cas9–Cas9 framework is not limited to the Cas9s utilized in this study, but that it should prove general and can be used with alternate Cas9s or perhaps even Cas12a[36], where the choice of the composite system (e.g. SaCas9 or NmCas9) will be dependent on the available PAMs within the desired target sequence and the orthogonality of the guide RNAs. Overall, RNA-programmable Cas9–Cas9 fusions offer superior features to independent nucleases: expanded targeting range, improved specificity, and efficient generation of uniform editing products.

Our analysis of the DNA repair products produced by Cas9–Cas9 fusions provides insights into methods to increase the efficiency of DSB-mediated genome sequence alterations by programmable nucleases. cNHEJ is typically the default choice for DSB repair in most stages of the cell cycle[30]. Precise ligation by cNHEJ of the ends of a DSB generated by a single nuclease restores the target site sequence, and in the context of an active nuclease leads to repeated cycles of cleavage and repair until imprecise repair occurs disrupting the target sequence. The predominant production of precise deletions after the generation of two synchronous blunt DSBs at a composite target site supports the cellular preference for cNHEJ for DSB repair. The ability of precise DNA repair to "mask" nuclease activity, which is normally evident only as sequence alterations resulting from imprecise repair, is evident from the dual nuclease experiments. This is particularly striking for dual nucleases at target sites where one of the Cas9 monomers is weakly active, such as at some of the Nm/SaCas9 target sites (TS5, TS7, and TS12; Supplementary Fig. 10) or at SpCas9 protospacers with suboptimal PAMs (Fig. 5a). DNA cleavage at these sites by an independent nuclease is inefficient, and since precise cNHEJ repair can restore the native DNA sequence, they are marked by low lesion rates. However, Cas9 nucleases employed in combination reveal the activity of the weak nuclease in the form of increased precise deletions between the cleavage sites. This distinction in repair end products between single nucleases and paired nucleases may underlie differences in the activity profiles of SpCas9$^{MT}$–dSaCas9 and SpCas9$^{WT}$–SaCas9$^{WT}$ nucleases at some target sequences (Figs. 1c and 4). The increase in indel rates for the fusion proteins over an independent nuclease could also be due in part to the relaxation of the local chromatin architecture as observed for proximal CRISPR targeting[37]. The one disadvantage of the production of efficient precise deletions by the dual nuclease system is that this system will likely prove less effective for homologous recombination than a single Cas9 nuclease that can potentially repeated cleave a site that is repaired precisely by cNHEJ.

One of the hurdles for therapeutic genome editing applications is the uncertainty of the functional activity of the lesions that are produced by individual nucleases when relying on imprecise DNA repair. At a therapeutically relevant site, Cas9–Cas9 dual nuclease fusions produce defined precise deletions comprising up to 97% of the modified genomes, which should produce specific alleles that have activities that can be defined in model systems prior to advancement in therapeutic applications. Cas9–Cas9 fusion-mediated production of precise deletions should be applicable to the development of therapeutic genome editing strategies for a number of genetic disorders. As the immediate extension of the results described in this study, efficient excision of the core regulatory elements within the *BCL11A* erythroid enhancer in CD34+ HSPCs via ex vivo delivery of Cas9–Cas9 ribonucleoproteins is likely to achieve higher rates of inactivation than the production of local indels by a single nuclease[21–23]. This increased efficiency should improve the therapeutic potency of the resulting cell product when autologously transplanted back into a patient for the treatment of β-hemoglobinopathies, such as sickle cell disease[24].

## Methods

**Plasmid constructs.** Our SpCas9-Sa/NmCas9 experiments employed the following plasmids: All sgRNAs are individually expressed under a U6 promoter from a pBluescript II SK(+)-based vector. All single-Cas9 or dual-Cas9 nuclease constructs are expressed via a CMV IE94 promoter from a pCS2-Dest gateway plasmid[15]. NmCas9 and SaCas9 open-reading frames for nuclease construction were obtained from Addgene (#48670 & #61591). The nuclease-dead versions of these constructs (dNmCas9: D16A, D587A, H588A and N611A; dSaCas9: D10A and N580A) are generated via site-directed mutagenesis. Sequences of representative SpCas9–SaCas9 and SpCas9–NmCas9 constructs used within this study are listed in Supplementary Data 5. These plasmids will be deposited to Addgene for community distribution. We used the single-strand annealing-based plasmid reporter assay developed by Porteus laboratory[26] to monitor nuclease activity. Nuclease target sequences are cloned into the M427 plasmid in between EcoRI and SbfI sites.

**Cell culture and transfection/electroporation.** The Human Embryonic Kidney (HEK293T), K562 and Jurkat cell lines were gifts from our collaborators M. Green, L. Castilla, and H. Göttlinger, respectively (all at UMass Medical School, Worcester, MA, USA). All three cell lines were authenticated by University of Arizona Genetics Core and tested for mycoplasma contamination at regular intervals. HEK293T were cultured in high glucose DMEM with 10% FBS and 1% Penicillin/ Streptomycin (Gibco) in a 37 °C incubator with 5% $CO_2$. The K562 and Jurkat cell lines were maintained in RPMI 1640 medium with 10% FBS and 1% Penicillin/ Streptomycin (Gibco) in a 37 °C incubator with 5% $CO_2$. We used cells at a passage number from 5 until 25 for transient transfection to assay nuclease activity. In 24-well format, about $1.6 \times 10^5$ cells were transfected by Polyfect transfection reagent (Qiagen) according to the manufacturer's suggested protocol. For Jurkat cells and K562 cells, $2 \times 10^5$ cells were used per electroporation using Neon® Transfection System 10 L Kit (Thermo Fisher Scientific) using the suggested electroporation parameters: Pulse voltage (1350 v), Pulse width (10 ms), Pulse number (3). For both single and dual nucleases we used 50 ng of each sgRNA-expressing plasmid and 50 ng mCherry-expressing plasmid, 50 ng of single nuclease (SpCas9 or NmCas9 or SaCas9) or 100 ng Cas9–Cas9 fusion expressing plasmid. In addition, pBluescript II SK(+) was added to the co-transfection mix to bring the total DNA mass to 300 ng per transfection. For the SSA-reporter assay, an additional 150 ng M427 reporter plasmid was added.

**GFP-reporter assay.** 48 h post-transfection cells are trypsinized and harvested into a microcentrifuge tube. Cells are centrifuged at 500×g for 2 min, washed once with 1× PBS, recentrifuged at 500×g for 2 min and resuspended in 1× PBS for flow cytometry (Becton Dickonson FACScan). 10,000 events were counted from each sample for FACS analysis. To adjust the transfection efficiency differences in between samples, cells were initially gated for mCherry-expression, and the percentage of EGFP-expressing cells (nuclease-positive events) were quantified within mCherry-positive cells. Experiments were performed in three replicates on different days. The data are reported as mean values with error bars indicating the standard error of the mean.

**Immunofluorescence.** HEK293T cells are transfected in six-well format via Polyfect transfection reagent (Qiagen) using the manufacturer's suggested protocol with 300 ng Cas9–Cas9 fusion expression plasmid and 150 ng of each sgRNA expression plasmid on a cover slip. 48 h following transfection, transfection media was removed, cells were washed with 1× PBS and fixed with 4% formaldehyde in 1× PBS for 15 min at room temperature. Following blocking (blocking solution: 2% BSA, 0.3% Triton X-100, within 1× PBS), samples were stained with mouse anti-hemagglutinin (Sigma, H9658, 1:500), and Alexa 488 donkey anti-mouse IgG (H +L; Invitrogen, A-21202, 1:2000), sequentially. VECTASHIELD mounting medium with DAPI (Vector Laboratories, H-1200) was used to stain the nuclei and to mount the samples on slide. Images were taken with Zeiss AxioPlan 2 IE Motorized Microscope System.

**Western blot analysis.** HEK293T cells were transfected with 50 ng of single nuclease (SpCas9 or NmCas9 or SaCas9) or 100 ng Cas9–Cas9 fusion expressing plasmid by Polyfect transfection reagent as described above. 48 h after transfection, cells are harvested and lysed with 100 µl RIPA buffer. 8 µl of cell lysate is used for electrophoresis and blotting. The blots are probed with anti-HA (1:2000; Sigma #H9658) and anti-β-actin (1:2500; Sigma # A5316) primary antibodies; then HRP-conjugated anti-mouse IgG (1:10,000; Abcam #ab6808) secondary antibody. Visualization employed Immobilon Western Chemiluminescent HRP substrate (EMD Millipore #WBKLS0100). The transfection and western blot experiments were performed in triplicate. Quantification of the western blot data were performed by measuring the intensity of the hybridization signals using the ImageJ analysis program[38].

**Target and off-target lesion analysis by deep sequencing.** Library construction for deep sequencing is modified from our previous report[15]. Briefly, 72 h after transfection, cells are harvested and genomic DNA extracted with GenElute

Mammalian Genomic DNA Miniprep Kit (Sigma). Genomic loci spanning the target and off-target sites were PCR amplified with locus-specific primers carrying tails complementary to the Truseq adapters (Supplementary Data 3). 50 ng input genomic DNA was PCR amplified with Q5 High-Fidelity DNA Polymerase (New England Biolabs): (98 °C, 15 s; 67 °C 25 s; 72 °C 20 s) ×30 cycles. For the construction of the UMI-based library, 50 ng input genomic DNA was first linearly pre-amplified with 10 nM final concentration 5p-BCL11A_enh58_UMI primer using the Q5 High-Fidelity DNA Polymerase (New England Biolabs): (98 °C, 60 s; 67 °C, 25 s; 72 °C, 20 s) × 10 cycles. To the same reaction mix, 500 nM final concentration 5p-DS_constant and 3p-BCL11A_enh58_DS primers were added for another round of amplification (98 °C, 60 s; 67 °C, 25 s; 72 °C, 20 s) for 30 cycles. Next, 0.1 µl of each PCR reaction was amplified with barcoded primers to reconstitute the TruSeq adaptors using the Q5 High-Fidelity DNA Polymerase (New England Biolabs): (98 °C, 15 s; 67 °C, 25 s; 72 °C, 20 s) x10 cycles. Equal amounts of the products were pooled and gel purified. The purified library was deep sequenced using a paired-end 150 bp Illumina MiSeq run with the exception of the *AAVS1* extended deletion analysis, which employed a paired-end 250 bp Illumina MiSeq run.

MiSeq data analysis was done with the help of Unix-based software tools. First, we employed FastQC[39] to determine the quality of paired-end sequencing reads (R1 and R2 fastq files). Next, we used paired end read merger (PEAR)[40] to pool raw paired-end reads and generate single merged high-quality full-length reads. Reads were then filtered according to quality via FASTQ[41] for a mean PHRED quality score above 30 and a minimum per base score above 24. After that, we used BWA (version 0.7.5) and SAMtools (version 0.1.19) for aligning each group of filtered reads to a corresponding reference sequence. To determine lesion type, frequency, size, and distribution, all edited reads from each experimental replicate were combined and aligned, as described above. Alignments were categorized into seven classes: unedited, SpCas9 indels, Nm/SaCas9 indels, precise deletions, imprecise deletions, SpCas9+Nm/SaCas9 indels, and inversions. Lesion types and frequencies were then cataloged in a text output format at each base using bam-readcount. For each treatment group, the average background lesion frequencies (based on lesion type, position and frequency) of the triplicate-negative control group were subtracted to obtain the nuclease-dependent lesion frequencies. Next, using R, a system for statistical computation and graphics[42], we assessed whether the Cas9–Cas9 fusions resulted in different lesion rates from two independent Cas9s. Percent of lesion rates were transformed using logit function followed by one-way analysis of variance (ANOVA) with the Randomized Complete Block Design (Supplementary Figures 5, 6, 10) or Completely Randomized Design (Fig. 6b). BH-adjusted *p*-values were calculated to counteract the problem of multiple comparisons of the data shown in Fig. 6b (ref. [43]). Statistical testing for the *AAVS1* extended deletion analysis (Supplementary Figure 7) was performed using a Student's *t*-test for comparison of the fraction of precise deletions for the independent and fusion nucleases.

For UMI analysis, we first used BWA (version 0.7.5) and SAMtools (version 0.1.19) for aligning each group of filtered merged-read pairs to a corresponding reference sequence ignoring the unique molecular barcodes. Next, we used a custom Python and PySAM script to process mapped reads into counts of UMI-labeled reads for each target, the mapped reads are filtered by setting mapping value larger than 30. Alignments were categorized into seven classes: unedited, SpCas9 indels, Nm/SaCas9 indels, precise deletions, imprecise deletions, SpCas9 +Nm/SaCas9 indels, and inversions. Next, we identified UMI duplicates and the minimal set of amplicons that can account for the full set of reads with unique UMIs. For each unique UMI, a minimum of four observations of the same sequence were required to consider the sequence to have a low likelihood of being an artifact (sequencing error in the UMI element). For those sequences meeting this threshold, the base sequences for a unique molecular barcode are consolidated to one read per molecule, and the reads with unique UMI were counted. The same pipeline was used separately for each sample with deletions spanning less than 50 bp. For the analysis of larger deletions in the *AAVS1* locus, we filtered raw reads according to their quality via FASTQ for a mean PHRED quality score above 30 and a minimum per base score above 24. These sequences were not filtered based on their mapping value, as the mapping score for large deletions is low. The resulting UMIs number tables were concatenated and loaded into GraphPad Prism 7 for data visualization.

**GUIDE-seq off-target analysis.** GUIDE-seq[12] was performed with the following adjustments to the original protocol. HEK293T cells were transfected using Polyfect transfection reagent (Qiagen) according to the manufacturer's suggested protocol with 50 ng of single nuclease (SpCas9[WT], NmCas9[WT], SaCas9[WT]) or 100 ng Cas9–Cas9 fusion expression plasmid, 50 ng of each sgRNA expressing plasmid, 50 ng of a mCherry expression plasmid and 5 pmol of annealed GUIDE-seq oligonucleotide. 72 h after transfection, genomic DNA was extracted with GenElute Mammalian Genomic DNA Miniprep Kit (Sigma) according to the manufacturer's suggested protocol. We prepared the GUIDE-seq library with the original adaptors according to protocols described by Joung and colleagues[12]. Each library was indexed within the P5 and P7 adaptors for multiplex sequencing. The libraries were deep-sequenced as a pool using two paired-end 150-bp Illumina MiSeq runs.

Deep sequencing data from the GUIDE-seq experiment was analyzed using the Bioconductor Package GUIDEseq (v1.4.1)[44]. The window size for peak aggregation was set to 50 bp. Off-target site identification parameters were set for SpCas9 as follows: min.reads = 2, min.reads.per.lib = 1, distance.threshold = 70, min.peak. score.1strandOnly = 2, upstream = 20, downstream = 20, max.mismatch = 6, PAM.pattern = "NNN$", allowed.mismatch.PAM = 2. For NmCas9, same parameters were used except the following: PAM.size = 8, PAM = "NNNNGHTT", PAM.pattern = "NNNNGNNN$", allowed.mismatch.PAM = 3, max.mismatch = 9. For SaCas9, same parameters were used as for SpCas9 except the following: PAM.size = 6, PAM = "NNGRRT", PAM.pattern = "NNGNNN$", allowed.mismatch.PAM = 3, max.mismatch = 6. The potential off-target sites identified for each nuclease are listed in Supplementary Data 2. The specificity ratio is calculated as the sum of the unique GUIDE-seq reads at the target site divided by all of the unique reads at all of the computationally identified off-target sites.

**Estimation of Cas–Cas9 target sites within human genome.** To identify potential SpCas9–SaCas9 and SpCas9–NmCas9 target sites across the human genome, we used a Perl regular expression to search for all possible target sites chromosome by chromosome using two sets of parameters: "canonical targets" SpCas9 PAM = NGG, SaCas9 PAM = GRRT, NmCas9 PAM = GATT, Site orientation = D1 or D2, spacing between edges of Cas9 target sites = 10–30 bp, or "expanded range" SpCas9 PAM = NNG or NGN, SaCas9 PAM = GRRT, NmCas9 PAM = GATT, Site orientation = D1 or D2, spacing between edges of Cas9 target sites = 10–100 bp. The number of potential target sites for SpCas9–SaCas9 and SpCas9–NmCas9 were compared to the number of potential sites for SpCas9 within the genome determined by the occurrence of an NGG PAM.

## Data availability

Illumina Sequencing data have been submitted to the Sequence Read Archive. These datasets are available under BioProject Accession number PRJNA496245. Statistical calculations are found in Supplementary Data 1. The authors declare that all other data supporting the findings of this study are available within the paper and its Supplementary Information files or upon reasonable request.

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

## Acknowledgements

We thank N. Rhind for the use of his FACS machine, E. Kittler and the UMass Medical School Deep Sequencing Core for their assistance with the Illumina sequencing. All new reagents described in this work are being deposited with the nonprofit plasmid-distribution service Addgene. This work was supported by the US National Institutes of Health (grant R01AI117839 to S.A.W. and J. Luban, grant 1R01GM115911 to S.A.W. and E.J.S., and grant R01HL093766 to S.A.W. and N. Lawson).

## Author contributions

M.F.B., P.L., and K.L., performed all cell-based experiments. P.L. and L.J.Z. performed the bioinformatic analysis. A.G., N.A., and S.F.K. assisted with nuclease construct generation. M.F.B., P.L., L.J.Z., E.J.S., and S.A.W. directed the research and interpreted experiments. M.F.B., P.L., and S.A.W. wrote the manuscript with input from all of the other authors.

## Additional information

**Competing interests:** The authors declare the following competing interests: The authors have filed patent applications related to genome engineering technologies; E.J.S. is a co-founder and advisor of Intellia Therapeutics; S.A.W. is a consultant for Beam Therapeutics.

