## [Peer Review File · Nature Communications]

Reviewers' comments:

Reviewer #1 (Remarks to the Author):

The authors describe and optimize a framework for orthogonal Cas9 chimeras as a simplifying improvement over earlier fusions between attenuated Cas9 and programmable DNA binding domains. They test versions of Cas9-chimeras in either single or dual nuclease format, and characterize them for gene editing efficiency, specificity, and in the case of Cas9-Cas9 nucleases precise deletion efficiency. The main limitation of this approach is that the size of these orthogonal Cas9-Cas9 fusion proteins is too large to be encoded on either AAV or lentiviral vectors. Its practical utility, particularly for clinical applications, will therefore hinge on whether it can be effectively deployed in ribonucleoprotein format.

Major

1. The authors have determined a nuclease architecture and binding site configurations where robust activity are observed in the context of Cas9-dCas9, but it remains unclear what the overall limitations are on spacer length between the two nucleases for efficient activity. In figure 1b, the longest lengths in the D1 and D2 configurations remain relatively active. More precisely determining this range would also help inform targeting range calculations.
2. Are dual nuclease fusions required to achieve high-frequency precise deletion or could similar results could be achieved using SpCas9 guided by two gRNAs (particularly as RNPs)? Similarly, in Figure 2b, the proportion of precise deletions induced with simultaneous introduction of independent SpCas9 and NMCas9 is also relatively high. On the other hand, the aggregate summary plots shown in Figure 2d and 2e do support the argument that the chimeric orthogonal Cas9 fusions enhance precise deletion frequency; it would be helpful to see this data plotted individually to understand the magnitude and frequency with which this enhancement is observed.
3. Other parameters that remains unclear regarding the Cas9-Cas9 chimeric fusions are the size range of precise deletions that can be made and whether target site orientation has an impact on precise deletion frequency. Presumably, this may be related to the optimal spacing for Cas9-dCas9 fusions as tested in figure 1b, but this should be separately and systematically tested in the dual nuclease context.
4. The authors test dual targets where one contains sub-optimal SpCas9 PAM and the other a perfectly matched SaCas9 PAM. Would cooperativity of binding permit usage of non-optimal PAMs at both binding sites?
5. In the case of a clinically relevant target such as the GATA1 binding site in the erythroid-specific enhancer of the BCL11A gene, it will be important to determine whether the Cas9-Cas9 chimeras can be used for precise deletion not only in HEK293T cells but also in relevant cell type of interest: human CD34+ cells. In this case, Cas9 orthogonal chimeric fusions would be ideally tested in ribonucleoprotein format. The practical utility of this approach really depends on whether it can be effectively deployed as an RNP.
6. In these GATA1 binding motif targeting experiments, it would be helpful to better understand the rationale for choice of targets containing suboptimal SpCas9 PAM sequences over those containing optimal SpCas9 PAMs in the vicinity. Is the increased targeting range required to target the GATA-1 site in BCL11A for precise deletion?
7. Based on GUIDE-seq data, the specificity of single nuclease orthogonal Cas9 chimeras is improved over wild-type Cas9. What is the effective targeting range of this platform? The ability to increase targeting range to a number of non-canonical PAM-containing targets is one of the major advantages, and it would be helpful to see an explicit, conservative targeting range calculation for

the human genome.

Minor

1. Why is a single-stranded annealing (SSA) plasmid reporter assay used for architecture and binding site configuration optimization, when most of the applications described in the manuscript rely on canonical NHEJ-mediated DNA repair?
2. The labelling of Figure 2b is confusing and should be clarified in the figure legend itself. Do the numbers above bars 3, 4, 6, and 7 indicate the percentage of mutations that are precise deletions?
3. In Figure 3d, the figure legend relating the colored bars to mutation category is missing.
4. In Figure 1f, there is a typo in the word 'Specificity'.
5. Discussion, p. 15. It is unclear how the authors conclude that the repair of Cas9-dSaCas9/dNmCas9 induced DSBs are repaired by non-canonical NHEJ? How can canonical versus non-canonical NHEJ repair products be distinguished without inhibition of the relevant repair factors such as DNA Ligase IV, etc.?
6. Is there a typo in the figure legend for Supplementary Figure 1b? Otherwise, this would appear to be redundant with Figure 1b.
7. In Supplementary Figure 2b and 2d, it would be helpful to add the percentage of precise deletions in relation to the total lesions, as they did in Figure 3c in the main text.
8. The final discussion point should be modified to reflect that there is a threshold of CD34+ cells (>2E6/kg) required for successful engraftment in stem cell transplantation, regardless of genome modification efficiency. While the number of cells administered will need to exceed this threshold, increasing genome editing efficiency should enhance the likelihood of observing a positive clinical benefit.

Reviewer #2 (Remarks to the Author):

In this manuscript, Bolukbasi et al. present a novel genome editing platform based on orthogonal CRISPR-Cas9 systems. They construct covalent fusions between the commonly-used SpCas9 and either SaCas9 or NmCas9 in various forms. Fusion of nuclease-dead versions of Sa or Nm decreases off-target mutagenesis by SpCas9, without undermining target cleavage efficiency. Because the fusions allow a range of separations between the recognition sites and they also work with SpCas9-MT, which is less fussy about its PAM, this enhancement should be effective at essentially any target recognized by SaCas9 or NmCas9.

Perhaps even more useful is the demonstration that fusion of active versions of SaCas9 or NmCas9 to active SpCas9 leads to efficient generation of precise deletions between the two predicted target sites. The authors demonstrate this outcome at several genomic targets in HEK293 cells, including the therapeutically-plausible BCL11A enhancer. This study makes a significant contribution to CRISPR-mediated genome editing and will find uses in many labs.

While improving some Cas9 editing features, these Cas9-Cas9 fusions may interfere with some others. The finding of very high proportions of precise deletions with two active Cas9's is somewhat surprising, since SpCas9 is known to leave 1- or 2-nt 5' overhangs at readily detectable frequencies, leading in particular to 1-bp insertions at some targets. The absence of deeper

deletions suggests that 5'->3' resection is inhibited at the dual cut sites, perhaps simply due to rapid NHEJ after simultaneous cleavage. Since resection is required for HDR, the paired nucleases may not be useful for this purpose, as suggested on p. 14. This should at least be tested. In addition, the very large size of the dual proteins (2580 aa; 7.74-kb cds) will challenge some delivery methods, especially some viral vectors.

I think the text would benefit from several clarifications:

- Say a little more about the plasmid assay used in Fig. 1 and elsewhere, so readers will know why GFP+ is the expected positive outcome.
- Discuss in the text the dependence of activity on spacer lengths (Fig. 1b). The fact that quite a range is tolerated is a very important feature of the system, since it allows PAMs for the different Cas9's to be located at variable distances from each other. The authors exploited this themselves in targeting genomic sequences.
- Say something about the length of the linker in the protein, even if others haven't been tested. It seems to be 62 aa, including the various NLS and HA sequences, and would likely be unstructured. Is cleavage between the Cas9's seen in cells?
- I'm sure readers would be interested in at least a brief discussion of reasons to prefer SaCas9 or NmCas9 in fusions with SpCas9. Perhaps neither is significantly better than the other. Data from the two appear to be combined in many places.
- The authors might also say a little something, even if just in the figure legend, about the differences among Sa sgRNAs in Fig. S2. Is this a reflection of the spacer lengths? It doesn't seem to correlate with the individual activities of the sgRNAs.

Minor points:

1. I couldn't figure out the difference between the histograms in Fig. 1b and Fig. S1b, although the %GFP values were different.
2. The legend to Fig. S1c should say what it shows – mostly cytoplasmic signal in the second and sixth vertical panels, nuclear in the others.
3. The images in Fig. S1c appear to show that the transfection efficiency is surprisingly low, around 20%. This seems incompatible with the high editing efficiencies reported in most experiments.
4. In the statement near the top of p. 10 that the targeting density should be the sum of those for SaCas9 and NmCas9, the authors should compare this number to that of SpCas9 alone.
5. The sequences in Fig. S6 would be easier to read if they were aligned by the GATA1 site.
6. It was hard to read the supplementary tables, since they weren't labeled and they didn't have legends.

We wish to thank the reviewers for their time, constructive criticism and suggestions for improvements to this manuscript. We believe that we have addressed their critical concerns in this revision, and that the resulting manuscript is better as a consequence. Below are the point-by-point responses to the reviewers' concerns in **Bold**. All changes to the main text in the revised version are highlighted in **yellow**. Note: the main text figures have increased from four to five, as we have broken Figure 1 into two different figures with the addition of new data. Likewise the number of supplemental figures has increased.

Reviewers' comments:

Reviewer #1 (Remarks to the Author):

The authors describe and optimize a framework for orthogonal Cas9 chimeras as a simplifying improvement over earlier fusions between attenuated Cas9 and programmable DNA binding domains. They test versions of Cas9-chimeras in either single or dual nuclease format, and characterize them for gene editing efficiency, specificity, and in the case of Cas9-Cas9 nucleases precise deletion efficiency. The main limitation of this approach is that the size of these orthogonal Cas9-Cas9 fusion proteins is too large to be encoded on either AAV or lentiviral vectors. Its practical utility, particularly for clinical applications, will therefore hinge on whether it can be effectively deployed in ribonucleoprotein format.

We agree that one critical stage in the future development of this system is the creation of purified Cas9-Cas9 fusion protein that can be delivered in ribonucleoprotein format to cells via nucleofection or another methodology. We have successfully generated these constructs and preliminary tests in CD34+ HSPCs demonstrate substantial editing. However, we believe that the description of the generation of this protein and its use *ex vivo* in primary cells is beyond the scope of this manuscript, which is focused on the initial description and validation of the Cas9-Cas9 fusion system.

Major

1. The authors have determined a nuclease architecture and binding site configurations where robust activity are observed in the context of Cas9-dCas9, but it remains unclear what the overall limitations are on spacer length between the two nucleases for efficient activity. In figure 1b, the longest lengths in the D1 and D2 configurations remain relatively active. More precisely determining this range would also help inform targeting range calculations.

We have employed the SpCas9^{MT3}-dSaCas9 nuclease at the AAVS1 locus with multiple different distances between the Cas9 target sites to assess the distance dependence of the synergy that is achieved in the linked Cas9 system for the D1 & D2 orientations. These data indicate that the fusion protein provides enhancement in the editing rates over a distance of up to 200 bp between the Cas9 binding sites (Fig 1c). Beyond this distance there is little evidence of enhanced editing activity.

2. Are dual nuclease fusions required to achieve high-frequency precise deletion or could similar results could be achieved using SpCas9 guided by two gRNAs (particularly as RNPs)? Similarly, in Figure 2b, the proportion of precise deletions induced with simultaneous introduction of independent SpCas9 and NMCas9 is also relatively high. On the other hand, the aggregate summary plots shown in Figure 2d and 2e do support the argument that the chimeric orthogonal Cas9 fusions enhance precise deletion frequency; it would be helpful to see this data plotted individually to understand the magnitude and frequency with which this enhancement is observed.

An aggregate plot of 41 individual target sites was provided in the original manuscript in Supplementary Figure 3C. This is now Sup figure 4C in the revised manuscript. To provide more detailed information on the changes in precise deletion rates for the fusion protein relative to the dual independent nucleases at each locus, we have added plots on the precise deletion rates for the 41 and 12 individual sites in Supplementary Fig 4D and 5F, respectively. These plots show substantial improvement for the Cas9-Cas9 fusion proteins relative to the dual independent nucleases at nearly every site.

3. Other parameters that remains unclear regarding the Cas9-Cas9 chimeric fusions are the size range of precise deletions that can be made and whether target site orientation has an impact on precise deletion frequency. Presumably, this may be related to the optimal spacing for Cas9-dCas9 fusions as tested in figure 1b, but this should be separately and systematically tested in the dual nuclease context.

An aggregate plot of data for 6 genomic target sites in each orientation (D1 & D2) was provided in the original manuscript in Supplementary Figure 4B & C. Based on this data there may be somewhat more nuclease activity in the D1 orientation than the D2 orientation, but both are highly active. To provide more detailed information on the changes in precise deletion rates for the fusion protein as a function of the distance between the target sites, we have added plots on the precise deletion rates for the 41 and 12 individual

sites in Supplementary Fig 4D and 5F, respectively, that are organized by distance between the target sites. These plots show that the Cas9-Cas9 fusion proteins are relatively agnostic to the distance between the sites for up to 30 bases of separation. Based on the AAVS1 activity SpCas9^{MT3}-dSaCas9 data for different spacings in Figure 1C, we believe that enhanced deletions can be made over distances of at least 100 bp.

4. The authors test dual targets where one contains sub-optimal SpCas9 PAM and the other a perfectly matched SaCas9 PAM. Would cooperativity of binding permit usage of non-optimal PAMs at both binding sites?

This is a very interesting question. We have utilized suboptimal PAMs for both nucleases at GATA1-TS5 [nAG & n4GTTT instead of nGG and n4GTTT], GATA1-TS10 [nAG & n2GGGC instead of nGG and n2GRRT], and GATA1-TS12 [nAG & n2GAGG instead of nGG and n2GRRT] (Supplementary Figures 7 & 8). For all of these target sites we see an enhancement in the fraction of precise deletions produced by SpCas9^{WT}-SaCas9^{WT} relative to the dual independent nucleases. The reviewer is correct that this is an interesting avenue to investigate, but we believe that it is beyond the scope of the initial descriptive manuscript on the fusion system.

5. In the case of a clinically relevant target such as the GATA1 binding site in the erythroid-specific enhancer of the BCL11A gene, it will be important to determine whether the Cas9-Cas9 chimeras can be used for precise deletion not only in HEK293T cells but also in relevant cell type of interest: human CD34+ cells. In this case, Cas9 orthogonal chimeric fusions would be ideally tested in ribonucleoprotein format. The practical utility of this approach really depends on whether it can be effectively deployed as an RNP.

We agree that this is an important aspect of this system to study for its future therapeutic application. As we noted above, we are working to develop the RNP for the Cas9-Cas9 fusions. Just the construction and purification of the Cas9-Cas9 fusion protein is a serious effort, which we believe belongs in a separate manuscript. We have evidence from our collaboration with the laboratory of Dan Bauer that Cas9-Cas9 RNPs can efficiently edit CD34+ HSCs, data from which was presented at the recent Keystone and ASGCT conferences on genome editing. This work will form the foundation of a future manuscript on the subject. To begin to address the functionality of the Cas9-Cas9 system in alternate cell lines, we provide data on the GATA1-TS9 target site in K562 and Jurkat cells delivered as plasmid expression systems via nucleofection. We observe enhanced editing and higher

precise deletion rates by the SpCas9^{WT}-SaCas9^{WT} relative to the dual independent nucleases (Supplementary Figure 10), which demonstrates that this is not just a HEK293T cell phenomenon.

6. In these GATA1 binding motif targeting experiments, it would be helpful to better understand the rationale for choice of targets containing suboptimal SpCas9 PAM sequences over those containing optimal SpCas9 PAMs in the vicinity. Is the increased targeting range required to target the GATA-1 site in BCL11A for precise deletion?

We now have a better perspective on the range that can be spanned by Cas9-Cas9 fusions, but at the time of the design of these experiments in the BCL11A locus, we were only confident that modest distances (30 bp max) between the target sites could be accommodated, which limited our choice of target sites due to the limited number of SaCas9 and SpCas9 PAMs surrounding this region. In Supplementary Figures 7 & 8, we did attempt canonical [NGG PAM] SpCas9 target sites with neighboring NmCas9 target sites that span the GATA1 binding site [TS1, 2, 3 & 4]. In these instances the NmCas9 target site was inactive, leading to low precise deletion rates. Given our original perception on distance restriction (<30 bp gap), we examined other SpCas9 PAMs within this region neighboring alternate NmCas9 and SaCas9 target sites, where the suboptimal PAMs were the only choice. With our current knowledge of the spacing flexibility of the Cas9-Cas9 system, we could in the future explore alternate SpCas9 nGG PAMs as well.

7. Based on GUIDE-seq data, the specificity of single nuclease orthogonal Cas9 chimeras is improved over wild-type Cas9. What is the effective targeting range of this platform? The ability to increase targeting range to a number of non-canonical PAM-containing targets is one of the major advantages, and it would be helpful to see an explicit, conservative targeting range calculation for the human genome.

As recommended by the reviewer, we have computationally calculated the potential targeting range of the Cas9-Cas9 fusions within the human genome at two different stringencies: optimal PAMs for both nucleases and conservative distance constraints (10 to 30 bp spacing depending between the target sites; “canonical”), and relaxed PAM for SpCas9 with an optimal Sa/NmCas9 PAM and relaxed distance constraints (10 to 100 bp maximum depending on the orientation “expanded”) (Supplementary Table 4, Supplementary Figure 6B). Reassuringly, even the conservative parameters indicate that there are only about 3-fold fewer target sites for SpCas9-SaCas9 fusions than

SpCas9 alone, and that for the optimistic parameters there are actually 2.5-fold more target sites for SpCas9-SaCas9 fusions than SpCas9 alone. Thus, the Cas9-Cas9 fusion system does expand the targeting range for sequences that can be cleaved by SpCas9 genome-wide.

Minor

1. Why is a single-stranded annealing (SSA) plasmid reporter assay used for architecture and binding site configuration optimization, when most of the applications described in the manuscript rely on canonical NHEJ-mediated DNA repair?

The SSA plasmid system allows the straightforward measurement of cleavage rates for the same protospacers with different orientation and spacing. Creating a series of genomic system that would allow the comparison of common guides with different spacings and orientations would be time consuming (e.g. a traffic light system).

2. The labelling of Figure 2b is confusing and should be clarified in the figure legend itself. Do the numbers above bars 3, 4, 6, and 7 indicate the percentage of mutations that are precise deletions?

This is defined in the figure panel itself – “% above the bar = precise deletion/total lesion”. In the revised manuscript we have added this to the legend (now fig 3B) as well.

3. In Figure 3d, the figure legend relating the colored bars to mutation category is missing.

Thank you for noting this oversight. We added a legend to the figure panel and a description to the figure legend.

4. In Figure 1f, there is a typo in the word ‘Specificity’.

Thank you for noting this error. We corrected this in the figure (now 2D).

5. Discussion, p. 15. It is unclear how the authors conclude that the repair of Cas9-dSaCas9/dNmCas9 induced DSBs are repaired by non-canonical NHEJ? How can canonical versus non-canonical NHEJ repair products be distinguished without inhibition of the relevant repair factors such as DNA Ligase IV, etc.?

The Reviewer is correct. We do not have direct evidence for the DNA repair pathways that are being utilized. We have modified this

discussion point to be more agnostic about the type of DNA repair that is being used.

6. Is there a typo in the figure legend for Supplementary Figure 1b? Otherwise, this would appear to be redundant with Figure 1b.

As indicated in the figure legend for Supplementary Figure 1b – these data are the inverted framework (e.g. SaCas9-SpCas9) – where the order of the nucleases is reversed (not SpCas9-SaCas9). Obviously, our writing was unclear on this point as both reviewers had the same concern. We reworked the main text to better alert the reader to this fact.

7. In Supplementary Figure 2b and 2d, it would be helpful to add the percentage of precise deletions in relation to the total lesions, as they did in Figure 3c in the main text.

We have made the requested adjustment (Now sup Fig 3B & D).

8. The final discussion point should be modified to reflect that there is a threshold of CD34+ cells (>2E6/kg) required for successful engraftment in stem cell transplantation, regardless of genome modification efficiency. While the number of cells administered will need to exceed this threshold, increasing genome editing efficiency should enhance the likelihood of observing a positive clinical benefit.

The reviewer is absolutely correct. The precision of our language was poor. We have reworked the closing sentence to better capture this idea.

Reviewer #2 (Remarks to the Author):

In this manuscript, Bolukbasi et al. present a novel genome editing platform based on orthogonal CRISPR-Cas9 systems. They construct covalent fusions between the commonly-used SpCas9 and either SaCas9 or NmCas9 in various forms. Fusion of nuclease-dead versions of Sa or Nm decreases off-target mutagenesis by SpCas9, without undermining target cleavage efficiency. Because the fusions allow a range of separations between the recognition sites and they also work with SpCas9-MT, which is less fussy about its PAM, this enhancement should be effective at essentially any target recognized by SaCas9 or NmCas9.

Perhaps even more useful is the demonstration that fusion of active versions of SaCas9 or NmCas9 to active SpCas9 leads to efficient generation of precise deletions between the two predicted target sites. The authors demonstrate this outcome at several genomic targets in HEK293 cells, including the therapeutically-plausible BCL11A enhancer. This study makes a significant contribution to CRISPR-mediated genome editing and will find uses in many labs.

While improving some Cas9 editing features, these Cas9-Cas9 fusions may interfere with some others. The finding of very high proportions of precise deletions with two active Cas9's is somewhat surprising, since SpCas9 is known to leave 1- or 2-nt 5' overhangs at readily detectable frequencies, leading in particular to 1-bp insertions at some targets. The absence of deeper deletions suggests that 5'->3' resection is inhibited at the dual cut sites, perhaps simply due to rapid NHEJ after simultaneous cleavage. Since resection is required for HDR, the paired nucleases may not be useful for this purpose, as suggested on p. 14. This should at least be tested.

The reviewer is absolutely correct. The Cas9^{WT}-Cas9^{WT} fusions will be poor for the production of HDR outcomes. We intended this description to be associated with the Cas9-dCas9 fusions, where only one nuclease is functional. We have clarified this in our revised manuscript.

In addition, the very large size of the dual proteins (2580 aa; 7.74-kb cds) will challenge some delivery methods, especially some viral vectors.

The reviewer is correct, which is why we envision that these will be used as ribonucleoprotein complexes *ex vivo* (as noted in the Discussion) or through some alternate delivery modality.

I think the text would benefit from several clarifications:

- Say a little more about the plasmid assay used in Fig. 1 and elsewhere, so readers will know why GFP+ is the expected positive outcome.

We have expanded this description in the text as requested.

- Discuss in the text the dependence of activity on spacer lengths (Fig. 1b). The fact that quite a range is tolerated is a very important feature of the system, since it allows PAMs for the different Cas9's to be located at variable distances from each other. The authors exploited this themselves in targeting genomic sequences.

As suggested, we have expanded this description in the text in the context of the calculation of the target site density for the Cas9-Cas9

fusions. See Reviewer #1 major point 7 for further response.

- Say something about the length of the linker in the protein, even if others haven't been tested. It seems to be 62 aa, including the various NLS and HA sequences, and would likely be unstructured. Is cleavage between the Cas9's seen in cells?

We have included a brief mention of the linker length in the text as requested. We have performed a Western blot (Supplementary Figure 1D), which indicates that there is no evidence of cleavage occurring within this linker in HEK293T cells.

- I'm sure readers would be interested in at least a brief discussion of reasons to prefer SaCas9 or NmCas9 in fusions with SpCas9. Perhaps neither is significantly better than the other. Data from the two appear to be combined in many places.

We added a general statement to the Discussion to describe the potential of the Cas9-Cas9 system to utilize a variety of different Cas9 nucleases or even Cas12a components, where the choice of these components would be dependent on the available PAMs within the desired target region.

- The authors might also say a little something, even if just in the figure legend, about the differences among Sa sgRNAs in Fig. S2. Is this a reflection of the spacer lengths? It doesn't seem to correlate with the individual activities of the sgRNAs.

As requested, we expanded the text description in the figure legend (now sup fig 3). In many cases the overall SpCas9-SaCas9 fusion activity correlates with the SaCas9 activity at its target site, when a common SpCas9 site is used, but this also depends on the PAM that is present and the relative register of the target sites.

Minor points:

1. I couldn't figure out the difference between the histograms in Fig. 1b and Fig. S1b, although the %GFP values were different.

Please see above in the response to Reviewer #1 – Minor point #6.

2. The legend to Fig. S1c should say what it shows – mostly cytoplasmic signal in the second and sixth vertical panels, nuclear in the others.

We altered the legend to describe this as requested.

3. The images in Fig. S1c appear to show that the transfection efficiency is surprisingly low, around 20%. This seems incompatible with the high editing efficiencies reported in most experiments.

These images are from an early experiment where we had not yet optimized the transfection conditions for a new cell culture plate size. We were not worried about the overall transfection efficiency in these pilot experiments, just the localization of the fusion protein.

4. In the statement near the top of p. 10 that the targeting density should be the sum of those for SaCas9 and NmCas9, the authors should compare this number to that of SpCas9 alone.

See response to this request in the Reviewer #1 reply above. Major point #7

5. The sequences in Fig. S6 would be easier to read if they were aligned by the GATA1 site.

This figure layout was adjusted as requested (now sup fig 7).

6. It was hard to read the supplementary tables, since they weren't labeled and they didn't have legends.

We have added descriptors for the supplementary tables in the supplementary PDF file and added legends to the individual excel panels as requested.

Reviewers' comments:

Reviewer #1 (Remarks to the Author):

The authors have revised their manuscript describing orthogonal Cas9-dCas9 and Cas9-Cas9 chimeras. While the majority of my questions were reasonably addressed, there remain two major questions that require further clarification:

Major

1. The size range of precise deletions that can be made by Cas9-Cas9- chimeric fusions (originally major comment #3) remains unclear but is important to define to fully appreciate the capabilities of the system. Given the data from new Fig. 1c that SpCas9MT3-dSaCas9 fusions have activity with <200 bp of distance between gRNA target sites, what are the corresponding distances at which precise deletions can be stimulated by SpCas9-SaCas9? (This information would also improve the ability to accurately estimate the targeting range of the dual nuclease system.)

2. As previously mentioned (originally major comment #5), the usefulness of this method, particularly for clinical applications, largely centers on whether it can be efficiently deployed in ribonucleoprotein format. This is because the size of the proteins themselves preclude encoding and packaging in AAV or lentiviral vectors (although in theory they could also be delivered as mRNA). Determining whether Cas9-SaCas9 dual nuclease fusions are active as RNPs, at least in model cell lines, should be considered to be within the scope of the current manuscript as it has substantial bearing on its overall practical utility.

Reviewer #2 (Remarks to the Author):

The authors have responded well to the comments of the initial reviews, and the revised manuscript is improved as a result, with some remaining issues.

- I find the new data on spacer-dependence (Fig. 1C and Supp. Fig. 2) difficult to understand. It seems implausible that both targets can be simultaneously occupied when they are 200 bp or even 100 bp apart, since the linker between the constituent proteins is around 60 amino acids. The full extent of an unstructured amino acid is about 3Å, which is less than the separation between adjacent base pairs in DNA. How does the enhancement work at these distances? Is the DNA being bent?
- SaCas9 sgRNA #5 seems weaker than #6 (Supp. Fig. 2B), while their effect on SpCas9MT3 cleavage is reversed (Fig. 1C). I would be grateful for some discussion of this point.
- It is not correct to say (p. 9) that the deletion outcomes for Sp-Sa fusions are similar to those for the Sp-Nm fusions shown in Fig. 3B, since the former show a much more diverse range of outcomes with different Sa sgRNAs (Supp. Fig. 3D). Obviously, not all combinations are equivalent.
- Although the authors agree in their response to the original reviews that the WT-WT fusions might not be useful for homology-directed repair, they did not state this explicitly in the revised Discussion (p. 16), which they should do.

Minor comments.

1. In the first line of the abstract, "are" should be "is"; the subject of the sentence is "development".
2. On p. 4 near the bottom in parentheses, change "referred as" to "referred to as".
3. On p. 6, in the sentence about N-terminal fusions, cite Supplementary Fig. 1B. And in the legends to Fig. 1B and Supp. Fig. 1B, state explicitly that the former is for C-terminal and the

latter for N-terminal fusions. This confused both reviewers initially.

4. The sentence added on p. 6 about the Western blot interrupts the flow regarding dual NLSs, and its relevance is not spelled out. I would suggest either moving this thought to earlier in the paragraph (low level of editing could be due to poor nuclear localization exacerbated by less efficient expression of the fusion protein [see Western blot]) or leaving it in its current location with more explanation (since the fusions are expressed less efficiently [see Western blot], it was important to use the dual NLS architecture).

5. The whisker plots in Fig. 3E seem at odds with the data in Supp. Fig. 3D, where the percent precise deletion is as low as 23% for some WT-WT fusions. Evidently the VEGFA-TF3 data were not included in these plots.

6. Fig. 4D doesn't add anything to Fig. 4C, and it takes careful reading of the legend to understand what it is illustrating. With the percent precise deletion presented in panel C, panel D is not needed.

7. On pp. 14-15, state explicitly that off-target sites were also identified for TS10 (computationally?) and note that deep sequencing (Fig. 5B) showed very low levels of mutagenesis at the top 10 targets for TS7 and TS10.

We wish to thank the reviewers again for their time, constructive criticism and suggestions for improvements to our revised manuscript. We have made additional revisions to the manuscript and added additional data that we believe addressed the majority of their remaining concerns about our first revised manuscript. Below are the point-by-point responses to the reviewers' concerns for the revised manuscript in **Bold**. All changes to the main text in the second revised version are highlighted in cyan.

Note: we have added a figure panel of new data to Figure 3(f), we have removed Figure panel 4D as requested by Reviewer #2, and we have added one Supplementary figure (#6) of new data, which shifted the numbering of the downstream figures.

Reviewer #1 (Remarks to the Author):

The authors have revised their manuscript describing orthogonal Cas9-dCas9 and Cas9-Cas9 chimeras. While the majority of my questions were reasonably addressed, there remain two major questions that require further clarification:

Major

1. The size range of precise deletions that can be made by Cas9-Cas9- chimeric fusions (originally major comment #3) remains unclear but is important to define to fully appreciate the capabilities of the system. Given the data from new Fig. 1c that SpCas9^{MT3}-dSaCas9 fusions have activity with <200 bp of distance between gRNA target sites, what are the corresponding distances at which precise deletions can be stimulated by SpCas9-SaCas9? (This information would also improve the ability to accurately estimate the targeting range of the dual nuclease system.)

As requested by the reviewer we have performed the analysis of the precise deletion rate over a range of different distance lengths within the AAVS1 locus that correspond to the SpCas9^{MT3}-dSaCas9 data presented in Figure 1C that was added in the prior revision of the manuscript. These new data are presented in a panel in Figure 3(f) and a new Supplementary Figure (#6). The results of this analysis mirror the data in Fig 1C. The SpCas9^{WT}-SaCas9^{WT} fusions display increased precise deletion rates over a range of distances that span ~200 bp. Beyond that distance the activity of the Cas9-Cas9 fusions is slightly lower than the independent Cas9s, and the fraction of precise deletions is similar to the independent nucleases suggesting that the fused Cas9s are no longer able to cut the pair of sequences synergistically. Text describing this new data is introduced in page 10.

2. As previously mentioned (originally major comment #5), the usefulness of this method, particularly for clinical applications, largely centers on whether it can be efficiently deployed in ribonucleoprotein format. This is because the size of the proteins themselves preclude encoding and packaging in AAV or lentiviral vectors (although in

theory they could also be delivered as mRNA). Determining whether Cas9-SaCas9 dual nuclease fusions are active as RNPs, at least in model cell lines, should be considered to be within the scope of the current manuscript as it has substantial bearing on its overall practical utility.

As noted in our response at the first stage of reviews, we agree that the development of RNPs is an important aspect of this system to study for its future therapeutic application. As we noted in our prior response, we are working to develop the RNP for the Cas9-Cas9 fusions. Just the construction and purification of the Cas9-Cas9 fusion protein is a serious effort, which we believe belongs in a separate manuscript. We have evidence from our collaboration with the laboratory of Dan Bauer that Cas9-Cas9 RNPs can efficiently edit CD34+ HSCs, data from which was presented at the recent Keystone and ASGCT conferences on genome editing. This work will form the foundation of a future manuscript on the subject.

As noted by the Reviewer, RNPs are not the only potential pathway for the delivery of the Cas9-Cas9 fusion constructs. Nanoparticle-mediated mRNA delivery is another potential route for the delivery of these constructs that should not be size restricted (e.g. Yin, H. *et al.* Therapeutic genome editing by combined viral and non-viral delivery of CRISPR system components in vivo. *Nature biotechnology* 34, 328–333 (2016)). Thus, we believe that these data within the revised manuscript, which now spans multiple cell lines, is sufficiently compelling to support its communication to the scientific community.

Reviewer #2 Comments to Authors

The authors have responded well to the comments of the initial reviews, and the revised manuscript is improved as a result, with some remaining issues.

- I find the new data on spacer-dependence (Fig. 1C and Supp. Fig. 2) difficult to understand. It seems implausible that both targets can be simultaneously occupied when they are 200 bp or even 100 bp apart, since the linker between the constituent proteins is around 60 amino acids. The full extent of an unstructured amino acid is about 3Å, which is less than the separation between adjacent base pairs in DNA. How does the enhancement work at these distances? Is the DNA being bent?

We agree with the reviewer that our initial anticipation based on linear distance that can be spanned by the linker intervening the two Cas9 modules would restrict synergistic nuclease activity to distances less than 60 bp based on the persistence length of DNA. We have now added additional data at the request of Reviewer #1 (SpCas9^{WT}-SaCas9^{WT} data at the AAVS1 locus; **Figure 3f and a new Supplementary figure 6) demonstrating the synergistic cleavage of target sites**

can span about 200 bp, consistent with the SpCas9^{MT3}-dSaCas9 data in (Fig. 1C and Supp. Fig. 2). One interesting feature of Cas9 DNA binding that differs from “normal” DNA-binding domains is the generation of an R-loop structure. Because the formation of an R-loop disrupts the duplex DNA structure, it is likely that upon the binding of the “first” Cas9 that the DNA can be locally bent to facilitate target acquisition by the “second” fused Cas9. In addition, in the context of the nucleus - where there are a variety of other DNA-binding proteins and nucleosome formation - DNA may not behave like a linear “rod”. In all honesty, we do not have an experimental “grounded” explanation for these results. We are satisfied that there is evidence of distance-dependence – it is just longer than we would have anticipated. We would be concerned if there was no distance-dependent behavior in the activity of these fusion proteins based on our hypothesized mechanism of increased local concentration.

- SaCas9 sgRNA #5 seems weaker than #6 (Supp. Fig. 2B), while their effect on SpCas9^{MT3} cleavage is reversed (Fig. 1C). I would be grateful for some discussion of this point.

The reason for this discrepancy is unclear, but it may stem from the fact that DNA-binding rates and DNA cleavage activity for SaCas9 may not necessarily correlate. Similar phenomena is observed in Sup fig3d for sgRNAs 2 vs. 3 vs. 4 vs. 5. Nuclease activity via the creation of mutations is an indirect readout where the cleavage at one site imperfectly repaired. It is possible that some sequences are precisely repaired at a higher efficiency than others. Alternately, our new data on the activity of the SpCas9^{WT}-SaCas9^{WT} fusions (**Figure 3 and a new Supplementary figure 6**) at sites #5 and #6 correlate with the activity of the SaCas9 sgRNAs (the independent nuclease activity and the Cas9 fusion nuclease activity correlates) unlike the SpCas9^{MT3}-dSaCas9 data. This latter data suggests that SpCas9^{MT3}-dSaCas9 data is not always a perfect surrogate for the prediction of the dual nuclease activity. We have added a line at **the end of p17 to note this discrepancy**.

- It is not correct to say (p. 9) that the deletion outcomes for Sp-Sa fusions are similar to those for the Sp-Nm fusions shown in Fig. 3B, since the former show a much more diverse range of outcomes with different Sa sgRNAs (Supp. Fig. 3D). Obviously, not all combinations are equivalent.

The Reviewer is certainly correct that not all combinations are equivalent. The data panels in Sup Fig 3D are organized based on distance and the orientation of the target sites, not based on the type of fusions (Sp-Nm vs Sp-Sa). We replotted the data from the 41 sites experiment sorted for the type of fusions (Sp-Nm vs Sp-Sa included below). As the reviewer can see from this graph, both types of fusions contain some variability in their activity, and as a consequence, we are reluctant to suggest that one is superior to another.

• Although the authors agree in their response to the original reviews that the WT-WT fusions might not be useful for homology-directed repair, they did not state this explicitly in the revised Discussion (p. 16), which they should do.

We have included this requested statement in the Discussion on p18.

Minor comments.

1. In the first line of the abstract, “are” should be “is”; the subject of the sentence is “development”.

Thank you – we have revised this sentence.

2. On p. 4 near the bottom in parentheses, change “referred as” to “referred to as”.

Thank you – we have revised this element.

3. On p. 6, in the sentence about N-terminal fusions, cite Supplementary Fig. 1B. And in the legends to Fig. 1B and Supp. Fig. 1B, state explicitly that the former is for C-terminal and the latter for N-terminal fusions. This confused both reviewers initially.

As requested, we have revised these paper components to emphasize the N-/C-terminal fusions.

4. The sentence added on p. 6 about the Western blot interrupts the flow regarding dual NLSs, and its relevance is not spelled out. I would suggest either moving this thought to earlier in the paragraph (low level of editing could be due to poor nuclear localization exacerbated by less efficient expression of the fusion protein [see Western blot]) or leaving it in its current location with more explanation (since the fusions are expressed less efficiently [see Western blot], it was important to use the dual NLS architecture).

This is an outstanding suggestion! We have moved the information on the Western blot analysis earlier in the paragraph as recommended by the Reviewer.

5. The whisker plots in Fig. 3E seem at odds with the data in Supp. Fig. 3D, where the percent precise deletion is as low as 23% for some WT-WT fusions. Evidently the VEGFA-TF3 data were not included in these plots.

Yes, as stated in the text (top of paragraph page 9) the 41 site dataset that was generated was performed to test the generality of the VEGFA-TS3 observation at different sites. We clarified the wording in this section to emphasize that these are additional target sequences. Thus, we excluded all of the VEGFA-TS3 sites in these datasets. Interestingly, for Sp-Sa fusions at TS3 sgRNA#2, where only 23% are precise deletions. The majority of the imprecise deletions (95%) are a single product (a single “C” insertion in the context of the expected deletion that suggests DNA cleavage at one of the nuclease sites is leaving a one base 5’ overhang in the context of these two guides).

6. Fig. 4D doesn’t add anything to Fig. 4C, and it takes careful reading of the legend to understand what it is illustrating. With the percent precise deletion presented in panel C, panel D is not needed.

The Reviewer is correct. We have removed panel Figure 4D as recommended.

7. On pp. 14-15, state explicitly that off-target sites were also identified for TS10 (computationally?) and note that deep sequencing (Fig. 5B) showed very low levels of mutagenesis at the top 10 targets for TS7 and TS10.

As requested by the Reviewer, we added two lines to the section describing the off-target editing rates and the information on the sites that included the computational prediction of off-target sites.

REVIEWERS' COMMENTS:

Reviewer #1 (Remarks to the Author):

The addition of data and figure panel satisfactorily addresses my first major question. It is tempting to speculate that the 200 bp limit represents the known distance around which DNA can easily bend into a 'circle' when wrapped around a nucleosome (~146 bp) plus the distance of the unstructured linker (~52 bp).

Demonstration that these nuclease chimeras can function when delivered in a non-plasmid format such as RNP or RNA would strengthen the manuscript's argument that this could be a useful platform for therapeutic precise deletion. However, I agree that the absence of this single point should not preclude its publication.

Reviewer #2 (Remarks to the Author):

I am satisfied with the authors' responses to the latest reviews. This study describes an original and useful addition to the Cas9 toolbox and deserves to be published.